# Environmental versus phylogenetic controls on leaf nitrogen and phosphorous concentrations in vascular plants

Di Tian [1,2,3] ✉, Zhengbing Yan[4], Bernhard Schmid [5,6], Jens Kattge [7,8], Jingyun Fang[6] & Benjamin D. Stocker[2,3,9,10] ✉

Global patterns of leaf nitrogen (N) and phosphorus (P) stoichiometry have been interpreted as reflecting phenotypic plasticity in response to the environment, or as an overriding effect of the distribution of species growing in their biogeochemical niches. Here, we balance these contrasting views. We compile a global dataset of 36,413 paired observations of leaf N and P concentrations, taxonomy and 45 environmental covariates, covering 7,549 sites and 3,700 species, to investigate how species identity and environmental variables control variations in mass-based leaf N and P concentrations, and the N:P ratio. We find within-species variation contributes around half of the total variation, with 29%, 31%, and 22% of leaf N, P, and N:P variation, respectively, explained by environmental variables. Within-species plasticity along environmental gradients varies across species and is highest for leaf N:P and lowest for leaf N. We identified effects of environmental variables on within-species variation using random forest models, whereas effects were largely missed by widely used linear mixed-effect models. Our analysis demonstrates a substantial influence of the environment in driving plastic responses of leaf N, P, and N:P within species, which challenges reports of a fixed biogeochemical niche and the overriding importance of species distributions in shaping global patterns of leaf N and P.

As fundamental elements for vascular plants on Earth, nitrogen (N) and phosphorus (P) are important in controlling photosynthesis, growth, and ecological functions of plants[1–4]. The relationship between leaf N and P concentrations is a key plant characteristic reflecting leaf economics[5–7] and has been used to interpret ecosystem nutrient limitation[8–11], carbon (C)- and N-cycle interactions under global change[12–14], or to link macroecology and biogeography with trait-based functional ecology[15–17]. Understanding the variation in leaf N and P

stoichiometric patterns and their underlying controls is therefore crucial for predicting responses of terrestrial ecology and biogeochemical cycles to environmental change.

Variations in leaf N and P stoichiometry along geographic gradients are pervasive[18–28]. For example, leaf N and P concentrations generally increase, but N:P ratios decrease from the equator to the cooler and drier mid-latitudes[22]. Such patterns have been interpreted in the light of different hypotheses regarding dominant controls.

[1]State Key Laboratory of Efficient Production of Forest Resources, Beijing Forestry University, Beijing 100083, China. [2]Institute of Agricultural Sciences, Department of Environmental Systems Science, ETH, Universitätsstrasse 2, 8092 Zürich, Switzerland. [3]Swiss Federal Institute for Forest, Snow and Landscape Research WSL, Zürcherstrasse 111, 8903 Birmensdorf, Switzerland. [4]State Key Laboratory of Vegetation and Environmental Change, Institute of Botany, Chinese Academy of Sciences, Beijing 100093, China. [5]Department of Geography, Remote Sensing Laboratories, University of Zürich, 8006 Zürich, Switzerland. [6]Institute of Ecology, College of Urban and Environmental Sciences, Peking University, Beijing 100871, China. [7]Max-Planck-Institute for Biogeochemistry, Hans-Knöll Street 10, 07745 Jena, Germany. [8]iDiv - German Centre for Integrative Biodiversity Research Halle-Jena-Leipzig, Puschstraße 4, 04103 Leipzig, Germany. [9]Institute of Geography, University of Bern, Hallerstrasse 12, 3012 Bern, Switzerland. [10]Oeschger Centre for Climate Change Research, University of Bern, Falkenplatz 16, 3012 Bern, Switzerland. ✉e-mail: tiandi@bjfu.edu.cn; benjamin.stocker@unibe.ch

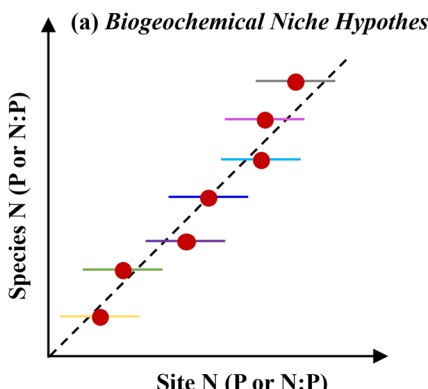
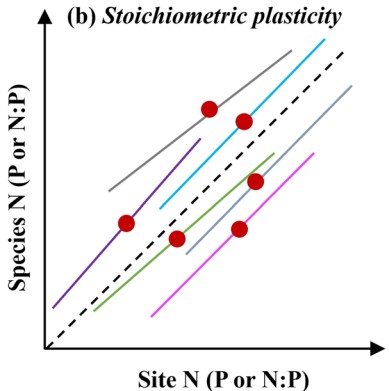
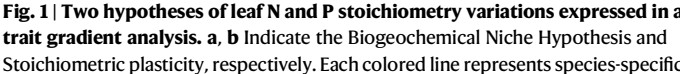

**Fig. 1 | Two hypotheses of leaf N and P stoichiometry variations expressed in a trait gradient analysis. a**, **b** Indicate the Biogeochemical Niche Hypothesis and Stoichiometric plasticity, respectively. Each colored line represents species-specific linear regressions of a species' leaf N (P or N:P) vs the mean leaf N (P or N:P) across all species occurring at the respective site.

Previous hypotheses have emphasized either the continental-scale variations of soil age and fertility[29]; the interactive effects of leaf N and temperature on biochemical reactions, including photosynthesis[30]; the effect of soil temperatures and chemistry on nutrient mineralization and availability to plants[22]; or the role of species distribution[21] in shaping global patterns of leaf N, P and N:P. Overall, these contrasting viewpoints provide contentious interpretations for global leaf N and P stoichiometry variation and take conflicting viewpoints regarding the importance of phylogenetic *vs* environmental factors in shaping these stoichiometric patterns.

Recently, studies[21,31] found an overriding effect of phylogeny and a vanishingly small influence of environmental variables on leaf nutrient concentrations based on linear mixed-effect models. Results were interpreted to support the importance of species distribution in shaping global-scale leaf nutrient patterns and the so-called "Biogeochemical Niche Hypothesis"[21]. This hypothesis posits that each species is characterized by a fixed leaf stoichiometry (low within-species variability) that matches environmental constraints and that geographical patterns in leaf nutrient distributions arise through environmental filtering of species occurrences. However, this interpretation of an overwhelming phylogenetic control on leaf N and P stoichiometry contrasts with previously documented influences of climatic and edaphic variables on leaf N and P stoichiometry[7,19,32].

The conflicting attributions of observed variation in leaf N and P stoichiometry to phylogenetic vs environmental variables are related to an inherent methodological challenge. The separation of these variables is usually undermined by their lack of independence. The distribution of plant species is largely driven by the abiotic environment[33,34]. Yet, species do occur over a certain range of environmental conditions. To what extent the environment drives phenotypic plasticity or genetic adaptation in leaf N and P stoichiometry also within species remains challenging to detect but is informative for testing the Biogeochemical Niche Hypothesis. Linear mixed-effect models (LMMs) have been widely employed for separating phylogenetic and environmental effects on leaf traits[21,35], motivated by their suitability to model structured data and their ability to control for phylogenetic effects and species identity as random terms, implicitly assuming that they are unrelated to the environment and given precedence over the latter in model fitting. More recently, tree-based statistical learning methods, for example, random forest models (RF), have been shown to be suitable for modeling leaf N and P[36,37]. These models, too, provide a natural way to simultaneously account for environmental (continuous) and phylogenetic (categorical) information. However, the implications of methodological choices for separating environmental vs phylogenetic variables so far have not been explicitly considered.

In view of these conflicting reports and methodological challenges, the question arises to what extent large-scale leaf N and P stoichiometric patterns are a reflection of different species (with their relatively fixed leaf nutrient stoichiometry) occurring at different sites along environmental gradients, and to what extent plasticity and genetic adaptation, driven by the environment, drive variation within species and contribute to large-scale patterns of leaf nutrient stoichiometry (Fig. 1).

To address this question, we compiled a global dataset of 36,413 paired leaf N and P concentrations per unit leaf mass of vascular plants, complemented by a comprehensive set of climatic, edaphic, and other environmental variables extracted from global datasets. Using these data, we first compared the power of linear regression models (LMs), LMMs, and RF models in explaining different components of variation in the data (variation across sites, within species, and across species). To enable comparability and facilitate the interpretation of our results in the context of the published literature, we specified LMMs to reflect the methodological choices of previous studies[21,31]. Then, we performed a trait gradient analysis (TGA)[38–40] to address the second question: does spatial variation in leaf N and P stoichiometry arise predominantly from species distribution and their respective stoichiometry (biogeochemical niche), or is there substantial within-species variation? We tested whether, according to the Biogeochemical Niche Hypothesis[21], the slopes of species-level trait gradient regressions were flat, as shown in Fig. 1. Alternatively, stoichiometric plasticity will yield positive slopes in the TGA. Perfect stoichiometric plasticity is indicated by slopes distributed around 1.

Our results show that variations within species are similarly strong as variations between species and that they are clearly influenced by the environment. These findings fill the gap of distinguishing and quantifying the role of species identity vs environmental controls on leaf nutrient stoichiometry and indicate that the previous Biogeochemical Niche Hypothesis, with its interpretation of an overriding and almost exclusive effect of phylogeny on leaf N, P, and N:P, should be revised.

## Results

### Variable selection and effects

We started by identifying the most important environmental variables for explaining variation in leaf N, P, and N:P. Reduced predictor sets, specific for leaf N, P, and N:P, respectively, enabled an improved model performance compared with models that included all 45 predictors (Fig. 2a–c) and were used for all subsequent analyses. In LMM models (Fig. 2d–f), N-deposition (ndep) had the strongest effect on leaf N and leaf N:P variation within species (both positive). The temperature of the coldest month (tmonthmin) had the strongest effect on leaf P

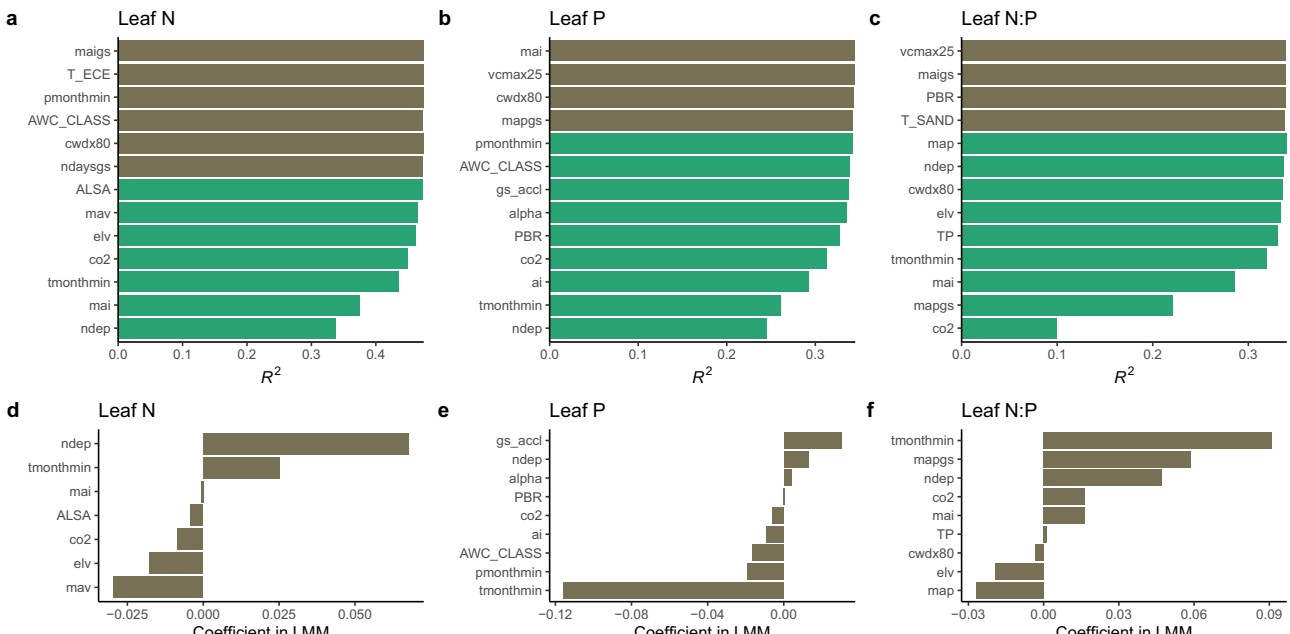

**Fig. 2 | Variable selection and effects. a–c** Variable selection order determined by recursive feature elimination based on Random Forest and 5-fold cross-validation for leaf N (**a**), P (**b**), and N:P (**c**). The last (and most important) variable to be removed in the recursive feature elimination is plotted at the bottom. The $R^2$ indicated by the bar at the bottom of panels **a–c** is for models with a single predictor ('ndep' for leaf N and P, and 'co2' for leaf N:P). The $R^2$ indicated by the next bar above is for a model with one additional predictor, as indicated by the label along the y-axis. The final selection of variables is indicated by the green bars. Brown bars indicate additional, next most important predictors, but not used for subsequent analyses. **d–f** Effect magnitudes of the selected variables, measured by the coefficients of normalized fixed effects in LMMs. Only variables for which the *t*-value in the respective LMM was significant at the 1%-level are shown. 'ndep' is nitrogen deposition, 'tmonthmin' is the mean temperature of the coldest month, 'ALSA' is the aluminum saturation of the soil solution, 'co2' is the atmospheric $CO_2$ concentration of the respective measurement year, 'elv' is elevation above sea level, 'mav' is the mean daytime vapor pressure deficit, 'gs_accl' is the predicted optimal stomatal conductance, 'ai' is the aridity index, 'pmonthmin' is the precipitation of the driest month, 'AWC_CLASS' is the available water storage capacity class, 'mapgs' is the mean growing season-total precipitation, 'map' is the mean annual precipitation. The remaining variable names are explained in Supplementary Tables 1–2. Source data are provided as a Source Data file.

(negative). Atmospheric $CO_2$ concentrations (co2), ndep, and tmonthmin were among the most important predictors for leaf N, P, and N:P. Soil variables were only selected among the most important variables for leaf N (aluminum saturation of the soil solution, ALSA) and for leaf P (soil texture, measured by the water holding capacity class, AWC_CLASS).

**Contrasting model performances**

The random forest (RF) models, fitted to site-level aggregated data (mean across all observations by site), with the selected subset of environmental variables as predictors, achieved an $R^2$ of 0.46, 0.34, and 0.34 for leaf N, P, and N:P, respectively, in contrast to an $R^2$ of 0.17, 0.19, and 0.19 in LMs (Fig. 3a, d, g). $R^2_{marg}$, measuring the proportion of variation explained by fixed (environmental) variables in LMMs, fitted to the full data, was only 0.04, 0.05, and 0.09 for leaf N, P, and N:P, respectively. In contrast, the proportions of variation explained by species identity were 0.68, 0.63, and 0.44 (intraclass correlation coefficient, ICC), respectively, for leaf N, P, and N:P ratio (Fig. 3c, f, i). When RF models were fitted to the full data, environmental variables explained a larger proportion of the variation than they did in LMMs, namely 0.13, 0.26, and 0.16 vs 0.04, 0.05, and 0.09, respectively. A similar contrast in the predictive power of environmental variables in RF and LMs is seen with models fitted to the modified data that contained only within-species variation (Fig. 3b, e, h). Here, RF models achieved an $R^2$ of 0.29, 0.31, and 0.22 for leaf N, P, and N:P, while LMs achieved an $R^2$ of 0.01, 0.03, and 0.07, respectively.

The power of environmental variables vs species identity in LMMs is subject to methodological aspects of the model-fitting procedure. By design, random factors are fitted with priority and "absorb" potentially shared effects with environmental variables. This is reflected also when comparing LMMs with LMs that are specified with distinct orders of the predictors (Supplementary Table 3). Subject to that order, species identity, and environmental variables contribute different sums of squares in LMs (Supplementary Table 3). Based on LMs, the shared effect of species identity and sites was dominant for leaf N, P, and N:P, explaining more than double of the variance explained by their separate effects (Supplementary Fig. 1).

**Trait gradient analysis**

The trait gradient analysis showed patterns similar to RF for leaf N, P, and N:P (Fig. 4). For most species, there was considerable variation of leaf N, P, and N:P within species across sites and this variation paralleled the variation in mean leaf N, P, and N:P of multiple species recorded at respective sites. The most frequent slope, i.e., the mode of the density distribution of slopes (Fig. 4d), is at 0.98 for leaf N, 0.80 for leaf P, and 1.01 for leaf N:P. A peak of the distribution of slope values close to unity indicates predominant plasticity or within-species genetic variation (henceforth referred to as 'intra-specific variation') of the respective trait along environmental gradients. However, for leaf N, a substantial number of species exhibited intra-specific variation with slopes <0.5. For leaf P, the most common degree of intra-specific variation appears somewhat smaller than the most common degree of intra-specific variation in leaf N and N:P. For leaf N, the distribution of slopes appears to be broader than for leaf P and N:P.

Ranges of site-level mean leaf N, P, and N:P, normalized by their respective overall mean, along which species occurred, tended to be the smallest for leaf N:P (median: 0.27), followed by leaf P (median: 0.31), and the largest for leaf N (median: 0.36, Fig. 4e). We found no correlation between normalized ranges and slopes, neither for N, nor for P, and no correlation between slopes for N and slopes for P (not

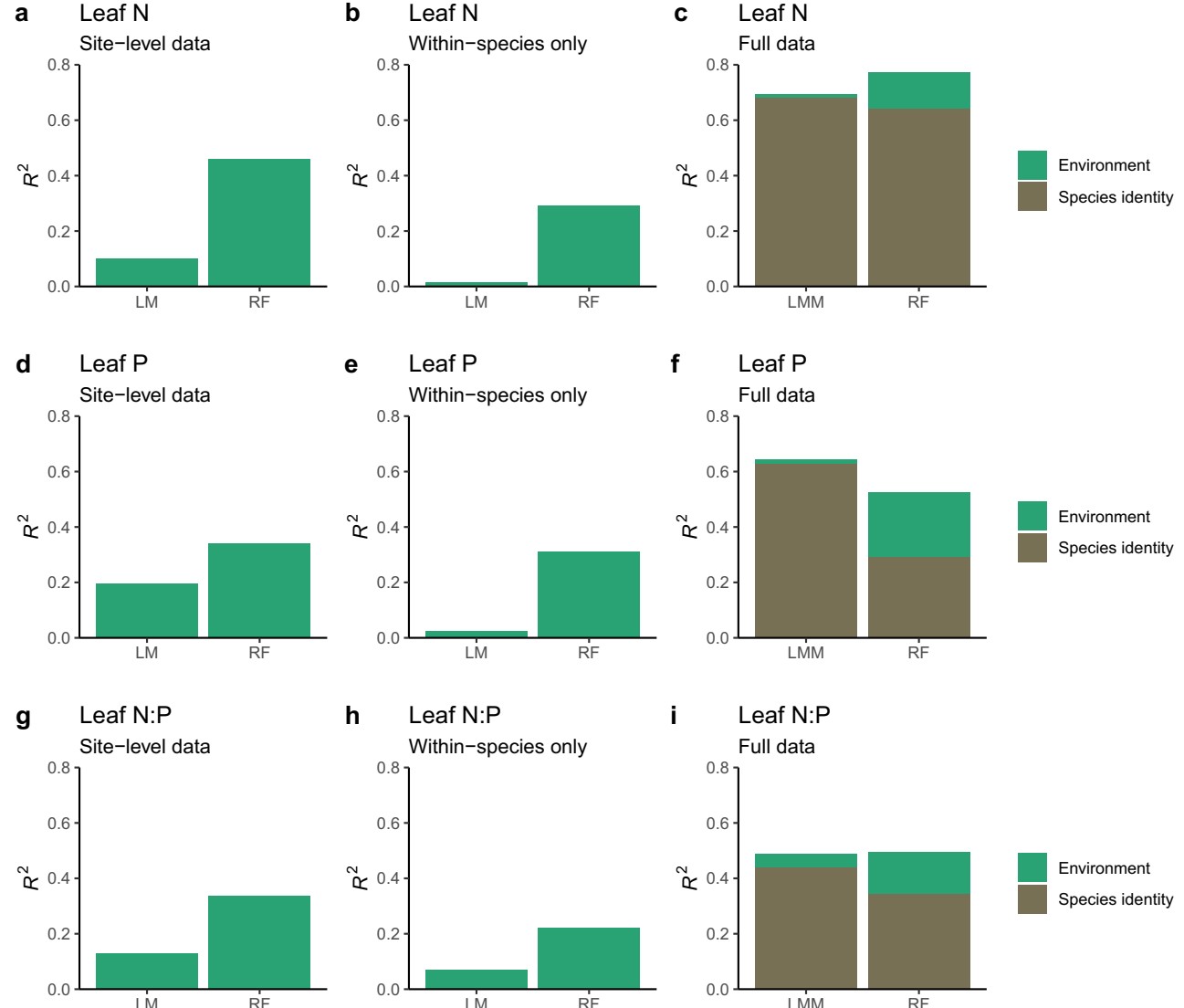

**Fig. 3 | Variations in leaf N, P, and N:P explained by species identity and environmental factors in random forest (RF), linear regression (LM), and linear mixed effect models (LMM).** **a**, **d**, **g** Proportion of variation in the data aggregated to the site-level explained by least squares regression models (LM) and RF. **b**, **e**, **h** Proportion of within-species variations explained by LM and RF models. **c**, **f**, **i** Proportion of variation in the full data explained by linear mixed-effect models (LMM) and RF models. The brown bars are determined as the intraclass correlation coefficient (ICC) from LMMs and as the cross-validation $R^2$ from RF models that contain only species identity, family, and genus information as predictors, but no environmental variables. The green bars on top of the brown bars represent the difference to the full models, where environmental variables and species identity were used as predictors. Source data are provided as a Source Data file.

shown). However, ranges in N and in P were positively correlated (Pearson's $r = 0.54$, Supplementary Fig. 3).

We found clear differences in the degree of intra-specific variation across species (Fig. 4f, g). For example, *Pinus sylvestris* and *Quercus petraea* appear to have relatively static leaf N, P, and N:P. In contrast, *Betula pendula* and *Picea abies* appear to be plastic (or exhibit intra-specific genetic variation) in leaf N, P, and N:P. Other species differ in their intra-specific variation between leaf N and leaf P. For example, *P. sitchensis* and *Q. ilex* have plastic (or genetically variable) leaf N, but relatively static leaf P. A substantial portion of species is "super-plastic", with slopes >1. This may reflect a strategy to overexpress a response to the environment through enhanced sensitivity to environmental variations than what would be expected from patterns in the site-level mean data.

### Other metrics of the importance of species identity

As indicated by the species variation decomposition, roughly half of the variations in leaf N and P arose within species, while the remainder was linked to variations across species (Fig. 4f). Among the three traits investigated, leaf N appeared to be most strongly linked to species identity, whereby interspecific variations explained 60% of overall variations ($R^2_{across} = 0.60$), while 40% of variations arose within species. The importance of species identity was weaker for leaf P ($R^2_{across} = 0.53$) and leaf N:P ($R^2_{across} = 0.42$).

## Discussion

We compiled a large dataset of leaf N, P, and N:P, paired with environmental covariates, and demonstrated that contrasting interpretations regarding the influence of environmental variables in driving leaf nutrient concentrations[7,19,21,22,24,31,32,41] reflect the particular structure in the data and methodological choices. The Biogeochemical Niche Hypothesis explains only half of the observed variations in leaf N, P, and N:P. Variations within species between sites are similarly strong as variations across species (Fig. 4f) and are strongly influenced by environmental variables. While certain statistical models succeeded at identifying them (e.g., RF), others did not (e.g., LMMs).

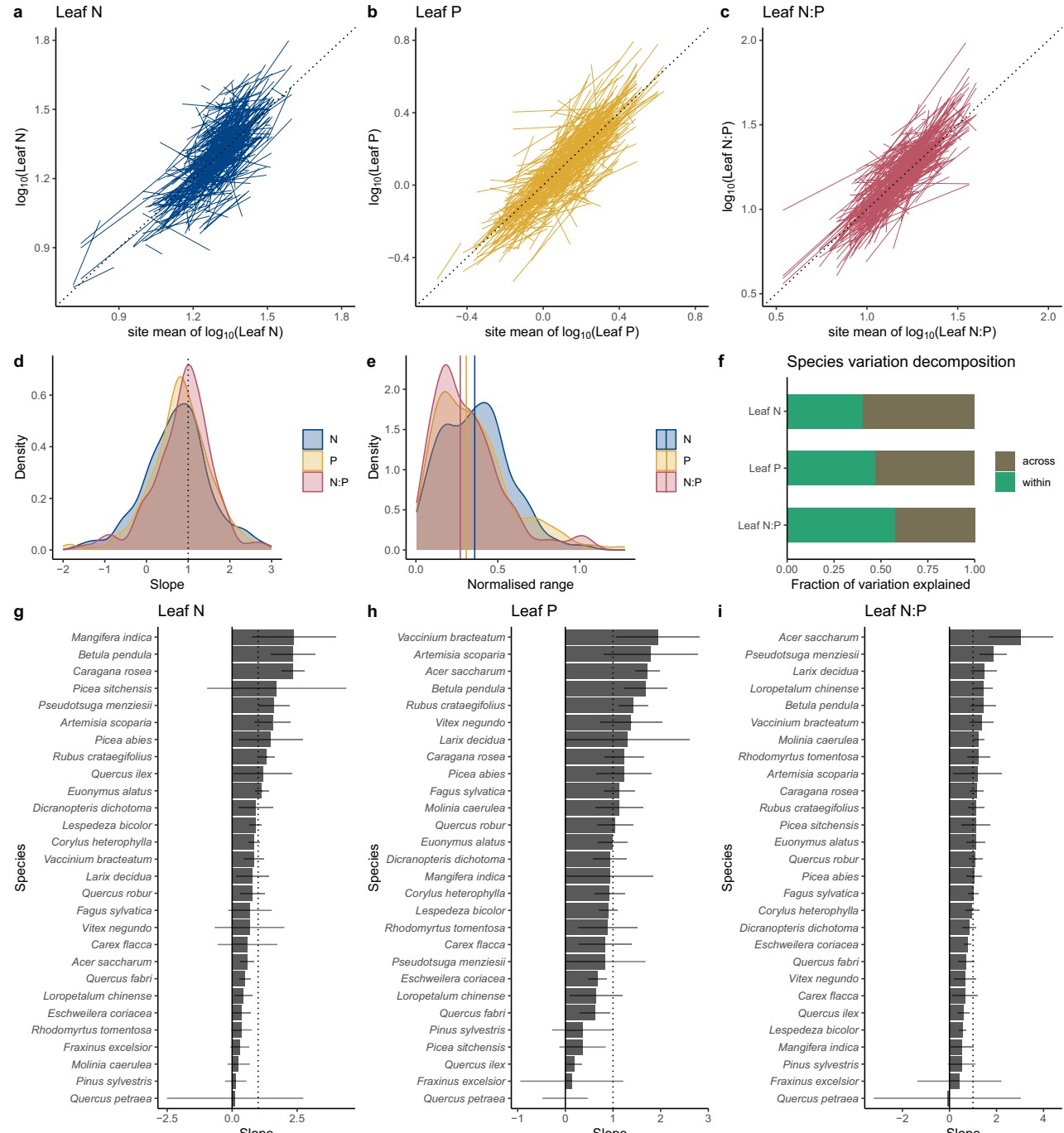

**Fig. 4 | Trait gradient analysis. a–c** Regression of log-transformed species-specific leaf N, P, and N:P versus the site-mean values (the dashed line shows the 1:1 relationship). **d** Distribution of slopes of species-specific regressions. **e** Distribution of species occurrence, quantified as the ranges of site-level mean leaf N, P, and N:P, normalized by their respective overall mean, along which species occurred. **f** Species variation decomposition (see "Methods" section), measuring the proportion of variations in leaf N, P, and N:P arising within vs across species. **g–i** Slopes of the species-specific regressions for species with the largest amount of data ($N > 50$). Gray lines indicate the 95% confidence interval of the slope estimate. Source data are provided as a Source Data file.

The influence of the environment as a driver of leaf N, P, and N:P is demonstrated by three results presented here. First, environmental variables explain 30–45% of variations in community-weighted means across sites (Fig. 3a, d, g). This reflects the environmental filtering of species occurrences across environmental gradients. Second, besides species distributions, the environment influences leaf N, P, and N:P directly, driving variation within species. RF explained around 20–30% of this variation, as shown in Fig. 3b, e, h. Third, a large proportion of species expressed strong plasticity in leaf N, P, and N:P, whereby

within-species variation paralleled across-site variation. This was expressed by the slope of species-specific regression lines in the trait gradient analysis (Fig. 4d).

Species occur over a substantial range of conditions, encompassing on average (median) 27–36% of the total range of site-mean leaf N, P, and N:P. Nevertheless, since species occurrence does not cover the full range of site-mean gradients, within-species plasticity explains only about half of the overall variation, suggesting an important but not overriding contribution of environmental filtering in

driving global leaf stoichiometry patterns. Nevertheless, it should be noted that through filtering species occurrence, the environment also drives variations of leaf N and P stoichiometry indirectly, i.e., among species. This effect plays out over time scales of species replacement (decades to centuries) and species evolution (centuries to millennia)[42,43], while intra-specific variation may be triggered by acclimation processes at time scales of weeks to years[44,45].

Taken together, these results suggest that the Biogeochemical Niche Hypothesis, according to which species with their distinct leaf stoichiometry occupy a narrow biogeochemical niche defined by the environment, falls short of explaining the "other half" of observed variation—the one arising from plastic responses of leaf N and P concentrations to the environment.

Our results here indicated that methodological choices and data structure lead to contrasting interpretations. Variation in leaf nutrient concentrations along large biogeographical gradients are well established and have been variably explained[16,19,22,46–51]. Empirical models of relationships between leaf nutrient concentrations and environmental variables explained 26%, 46%, and 55% in ref. 7, or 13%, 23%, and 19% in ref. 23 for leaf N, P, and N:P ratios, respectively − comparable to our results (46%, 34%, and 34%, respectively, for leaf N, P, and N:P by RF models). Among the strongest effects identified here was a positive effect of N-deposition on leaf N (consistent with refs. 51–55.) and N:P (consistent with refs. 10,56–58), a negative effect of temperature of the coldest month on leaf P, and a positive effect of the same on leaf N:P (consistent with refs. 22,26,59.). These apparently robust traits–environment relationships have given rise to spatial upscaling[31,60], mapping leaf traits with global coverage.

However, most global analyses reporting significant effects of environmental variables relied on aggregated data—either data aggregated to the site-level[19,23,47] or to biomes or ecoregions[7,46], or they relied on fitting separate models for a set of plant functional types and biomes[31]. This reduces the role of variability that arises across species and that remains unexplained by models of environmental controls. Indeed, Sardans et al. [21] reported negligible effects of environmental variables after the influence of species identity was removed. Although their findings do not necessarily undermine the validity of environment-based models for modeling spatial patterns of leaf traits, they do suggest that apparent large-scale trait–environment relationships arise almost exclusively as a reflection of environmental filtering and competitive selection of species with distinct leaf nutrient concentrations occupying their respective "biogeochemical niche"[21,51].

Our results suggest that this interpretation should be revised. The very small explanatory power of environmental variables reported in previous publications is linked to the limited capability of models (LMMs used in refs. 19,23,32,61) in fitting complex patterns in the data. There is a methodological challenge in attributing effects to environment vs species identity, arising from the fact that the two are correlated. The large shared effect between environment and species identity (Supplementary Fig. 1) cannot be decomposed without relying on targeted experimental designs and their interactions and non-linearities are not considered in published LMM-based analyses[19,21]. Our LMM model specification imitated their methodological choices. By design, in their LMMs the shared effect gets attributed to species identity if used as random-effects term, rather than to environmental variables used as fixed-effect terms in LMMs. Our exploration of alternative model formulations (Supplementary Table 3 & Fig. 1) demonstrates this and suggests that the interpretation of an overriding effect by species identity and a vanishingly small influence of environmental variables should be considered with caution and in the light of methodological limitations in separating respective effects. Random forest models learn interactive and non-linear effects of multiple environmental variables more effectively, detect trait–environment relationships beyond those arising from species composition (Fig. 3b, e, h), and yield superior results compared with

linear regression models in out-of-sample evaluations when fitted to leaf N and P data at hand (Fig. 3a, d, g). However, fitted model coefficients of RF are not always directly interpretable − in contrast to coefficients of LMMs. It has been argued before that "to the extent that the occurrence of species and environmental variation among sites are correlated, the two causes cannot be separated"[62]. We found here that, to the extent that within-species variations remain and reflect the influence of environmental variables, suitable statistical models may learn them.

Nevertheless, despite the large data volume and the extensive set of environmental covariates considered, less than half of the variation in the aggregated data can be explained even by the best models identified here and in previous studies[7,31]. This reflects a known challenge and has been interpreted as an expression of alternative leaf nutrient concentrations being maintained by different species for the same environmental conditions − potentially a consequence of equifinality of alternative functional trait combinations for competitive fitness[43,63].

In addition, limitations in the data are inevitable. Global datasets used for creating the predictors here and in previous studies (e.g., refs. 15,31,32) rely on limited information and assumptions for spatial upscaling. Data quality is limited also by the accuracy of the geolocation of individual records and by the fact that small-scale heterogeneity in the environment influences plant growth but is not captured by global datasets. In particular, soil maps are often unreliable at small scales and the influences of edaphic variables are complex[32,64]. When used for modeling in combination with climatic variables, the latter are often ascribed higher importance due to the limited reliability of global soil maps in capturing small-scale variations in soil quality[32] and to the inherent correlation between climate as a soil-forming variable and mapped soil types in global datasets. This likely also contributed to the results obtained here, where only one or no soil-related variable was selected for final models (Fig. 2a–f), while N-deposition and climatic variables consistently scored among the most influential variables. Notwithstanding the heterogeneity of the environment and the practically unavoidable data-quality limitations, other global trait variations have been predicted from the environment with more precision (e.g., photosynthesis traits in ref. 65, or area-based leaf nutrient concentrations in ref. 15). This suggests that fitness equifinality of functional traits and/or uncertain soil variables are particularly influential for mass-based leaf nutrient concentrations.

Knowledge about the role of environment vs phylogeny in controlling within- vs across-species variation in leaf nutrient concentrations is essential for predicting the impacts of global environmental change, simulating a temporal change in functional traits and leaf nutrient concentrations in terrestrial biosphere models, and the applicability of eco-evolutionary optimality-model concepts for leaf nutrient concentrations[66]. The treatment of stoichiometric flexibility is a key source of uncertainty in vegetation model predictions of responses to altered $CO_2$ and ecosystem nutrient inputs[67,68]. Rates of change in ecosystem-level trait averages are governed either by species turnover on a time scale of decades to centuries, or by acclimation of plant physiology within individuals to a changing environment on a time scale of weeks to years[69].

Leaf nutrient concentrations have been shown to respond at relatively short time scales within species exposed to experimental manipulation of the growth environment over a few years[55,70–72]. This is consistent with the considerable within-species variation found here and suggests that acclimation of leaf stoichiometry to decadal-scale climate change is an important response of plants that modifies global biogeochemical cycling, affects nutrient balances and limitations, and should be captured by mechanistic models in Earth system-change simulations.

A limitation of our study is that within-species variations may arise either from a plastic phenotypic response or from genetic

differentiation among disparate communities of the same species. This distinction is important for temporal modeling for the same reasons as described above, but the two processes cannot be discerned with the data used here. Nevertheless, results from ecosystem manipulation experiments suggest that phenotypic plasticity (or rapid evolution based on standing genetic variation within populations) is influential and that acclimation within individual plants plays an important role under global environmental change[73–77].

Global mean atmospheric $CO_2$ was among the most important selected variables for all three target variables and the single most important predictor for leaf N:P. The direction of the effect on leaf N identified here (decline with increasing $CO_2$) is consistent with observations from Free Air $CO_2$ Enrichment (FACE) experiments[70,78–80] and with the mechanistic understanding of the influence of $CO_2$ on N demand[81]. However, since we did not consider time or calendar year as a separate predictor, $CO_2$ may act in its role. Nevertheless, its influence in models lends further support for considerable plasticity in leaf nutrient concentrations over time, acclimating to temporal change in environmental conditions.

In conclusion, based on the most comprehensive global dataset of leaf N, P, N:P, and environmental covariates so far, our research provides a balanced assessment of the effects of environmental and phylogenetic controls on leaf N and P concentrations of terrestrial plants using novel methods of machine learning. The role of the environment in both filtering species and influencing traits within species, combined with the widespread use of linear mixed effects models with limited function of distinguishing statistical independence of phylogeny and environmental effects, predisposed published analyses to miss strong effects of environmental variables. We show that variations within species are similarly strong as variations between species and are clearly influenced by the environment. This indicates that the global pattern of leaf N and P stoichiometry is not merely driven by the distribution of plant species with their characteristic and fixed foliar N and P concentrations, but also reflects phenotypic plasticity or genetic adaptation to the growth environment. The finding of a clear environment-driven within-species variation is relevant for informing global vegetation models and their treatment of stoichiometric flexibility − a key source of uncertainty in their prediction of responses to a future environment. Our results suggest that current and future global environmental change can shift plant nutrient demand and ecosystem nutrient balances through the influence of a changing climate, $CO_2$, and N-deposition on leaf N, P, and N:P− even before leaf stoichiometry changes as a consequence of changing species compositions. This insight should be considered when predicting vegetation responses and the feedback between terrestrial biogeochemistry and global environmental change in Earth System Models.

## Methods
### Leaf N and P data
The dataset used here is extended from Tian D et al.[82] − a compilation of matched leaf N and leaf P concentrations (both in units of $mg\,g^{-1}$ based on dry leaf biomass) and mass-based leaf N:P measurements from the literature and large-scale field investigations in China[18,46,47,83,84], respectively, from the TRY Plant Trait Database[85], and from the International Co-operative Programme on Assessment and Monitoring of Air Pollution Effects on Forests, ICP Forests[21,86]. N:P values higher than $70\,g\,N\,(g\,P)^{-1}$ (0.04% of the data) were removed from the dataset. All sites had only one sampling date. In total, the dataset contains 36,413 individual records, collected from 7549 distinct sites and 3625 distinct species in 1383 genera and 203 families. Sites cover a wide climatic gradient and all major biomes (Supplementary Fig. 2). Species names are homogenized following Flora of China (http://frps.eflora.cn/), Useful Tropical Plants (http://tropical.theferns.info/), Australian Native Plants (https://www.anbg.gov.au/

index.html), Angiosperm Phylogeny Website (http://www.mobot.org/MOBOT/research/APweb/)[87], and The Plant List (www.theplantlist.org; accessed March 2021).

### Environmental data
All plots from which data were obtained were georeferenced (WGS84 standard). Using information on longitude, latitude, and elevation, we complemented our dataset by extracting data for a set of environmental variables from maps with global coverage. The geographical position of sites was verified and improved using original publications and Google Earth (https://earth.google.com/web/). To complement missing records of elevation, we extracted information for respective site locations from ETOPO1[88]. All data extraction was done using the *ingest* R package[89]. In total, we used 45 environmental variables, representing climatic, edaphic (soil-related), and other environmental variables (Supplementary Tables 1–2).

We computed 12 climatic variables (Supplementary Table 1) based on WATCH-WFDEI climate data[90], covering the years from 1979 to 2012, and down-scaled these climatic variables based on high-resolution WorldClim climatology[91]. In our dataset, sampling dates of leaf N and P spanned from 1935 to 2015. We used climatic time series here to account for long-term changes in climate in combination with known leaf N and P sampling dates. For sampling dates before 1979, we considered the mean climate for 1979–1988. Additionally, the annual mean ratio of actual over potential evapotranspiration (*alpha*), mean annual total evapotranspiration (AET), and the aridity index of annual mean precipitation over potential evapotranspiration (AI) were estimated using the SPLASH land water balance model and potential evapotranspiration based on the Priestly-Taylor Equation[92]. We additionally included model-based estimates of the maximum rate of carboxylation normalized to $25\,°C$ ($V_{cmax25}$), the electron transport for ribulose-1,5-bisphosphate (RuBP) regeneration ($J_{max25}$), and a multi-day average stomatal conductance ($g_s$). These were obtained from point-scale simulations of the P-model[93], using its implementation in the *rsofun* R package[94], predicting leaf-level acclimation of photosynthetic traits ($V_{cmax25}$, $J_{max25}$, $g_s$) based on optimizing the trade-off between carbon gain and water loss[95]. Due to the intrinsic[96] and widely observed[97] link between $V_{cmax25}$, Rubisco, and leaf N and P contents, we used these estimates here as predictors for leaf N, P, and N:P, thus attempting to account for simultaneous effects of multiple climatic variables (air temperature, VPD, elevation, irradiance, $CO_2$) on leaf nutrient stoichiometry. As an alternative aridity-related predictor, we included an estimate of the 80-year maximum cumulative water deficit ($CWD_{X80}$) from ref. 98. In total, we used 19 climate-related variables (Supplementary Table 1).

Soil properties related to texture and fertility were extracted from different digital global soil maps. Specifically, several properties related to soil structure, texture, and ion exchange capacity, as listed in Supplementary Table 2, were extracted from the Harmonized World Soil Database v 1.2 (HWSD, https://www.fao.org/soils-portal/soil-survey/soil-maps-and-databases/harmonized-world-soil-database-v12/)[99]. Aluminum saturation (ALSA), organic carbon content (ORGC), total nitrogen content (TOTN), and soil C:N ratios were extracted from the harmonized dataset of derived soil properties for the world (WISE30sec, https://data.isric.org)[100]. Soil phosphorus concentration using the Bray method (PBR), total phosphorus concentration (TP), and potassium (TK) concentration were extracted from The Global Soil Dataset for Earth System Modeling (GSDE, http://globalchange.bnu.edu.cn/research)[101]. To keep the soil properties comparable, the standard for soil layers was set around 30 cm (i.e., 0−30 cm for HWSD, 0−40 cm for WISE30sec, and 0−28.9 cm for GSDE).

Using information on site-specific sampling dates, we complemented the dataset with atmospheric $CO_2$ measurements from Mauna Loa Observatory[102], averaged over respective years, uniformly for all sites (assuming globally well-mixed concentrations). Also using

information of sampling dates, estimates of dry and moist atmospheric reactive N-deposition were extracted from outputs of global simulations of atmospheric chemistry over the historical period by ref. 103. Attempting to account for the small-scale redistribution of soil nutrients along the hillslope and local climatic effects mediated by the landscape position of the measurement plot, we included the Compound Topography Index (CTI)[104] as an additional predictor.

## Statistical models

We started our analyses by selecting a subset of the most important predictors for leaf N and P concentrations and leaf N:P using a recursive feature elimination. Relying on a limited set of predictors reduces collinearity among them and the potential for overfitting in subsequent model-based analyses. Starting from models that included all 45 available predictors, we iteratively removed the single predictor that led to the smallest decrease in the $R^2$ of observed versus modeled values, determined from a five-fold cross-validation. We retained a final set of predictors considering the $R^2$ determined on a five-fold cross-validation. The feature elimination was based on random forest (RF) models and was performed on data aggregated to site-level community-weighted mean leaf N and P concentrations, and leaf N:P. The algorithm's hyperparameter *mtry*, specifying the number of predictors considered at each split in individual decision trees, was set to $mtry = (K-1)/3$, where $K$ is the total number of predictors retained at the respective step of the feature elimination. The minimum node size (*min.node.size*, controlling the depth of decision trees) was set to 5 at all steps of the feature elimination and was chosen based on a prior hyperparameter search considering the root mean square error determined from a five-fold cross-validation. RF models were fitted using the *ranger*[105] and *caret*[106] libraries in R.

With the selected subset of predictors for each of the three target variables, we then performed a model-fitting comparison to cross-compare different modeling approaches (linear mixed models, LMM, and linear regression models, LM, *vs* RF) and the role of data aggregation. Models were fitted with the subset of selected predictors (i) to the full, species-level data, (ii) to data aggregated to the site-level considering community-weighted means of all variables, and (iii) to modified data that contained only within-species variations. For the latter, we modified leaf N and P concentrations, and N:P values by subtracting species-mean leaf N, P, and N:P values from all values of the respective species and considered these modified values as targets for modeling.

RF models were fitted to all three types of data. For RF models fitted to the full data (i), information about species, genus, and family identity was one-hot encoded. Hyperparameters *min.node.size* and *mtry* were tuned for each model separately, considering the mean root mean square error determined across five cross-validation folds. Reported $R^2$ values were quantified as the mean across five cross-validation folds. We quantified the proportion of variation in the full data explained by environmental variables alone by taking the difference between RF models fitted with environmental variables and species, genus, and family identity as predictors, and RF models with only species, genus, and family identity as predictors.

LMMs were fitted to the full data with the subset of environmental predictor variables as fixed effects, considering species identity as a grouping variable for random intercepts, and using the *nlme* library[107] in R. Before fitting LMMs, data were Yeo-Johnson-transformed. Partitioning of the variation explained by environmental variables (fixed variables) vs species identity (grouping factor for random offsets) was done using the R package *performance*[108]. We quantified the marginal $R^2$ ($R^2_{marg}$) to estimate the proportion of variation explained by the fixed effects − representing environmental variables. The intraclass correlation coefficient (ICC) was calculated to quantify the proportion of variation explained by species identity. For data aggregated to the site-level and the modified data that contained only within-species variation, we fitted linear regression (ordinary least squares) models

(LM), using the same subset of selected predictors. Note that LMs were fitted to those data because across-species variations, accounted for by random factors in LMMs, are removed by design at the site-level aggregated data and in the modified data that contains only within-species variations.

We additionally performed an analysis of variance (ANOVA) of species identity and environmental variables, investigating its dependency on including species as a random factor in LMMs and on the order of specifying species as a "fixed" factor in LMs. This was done by comparing ANOVA tables resulting from fitting species identity as fixed effect before or after environmental variables, using the *lm* function in base-R. Additionally, we separated individual and shared effects of site and species based on linear models where the respective predictors were fitted in different orders (Supplementary Fig. 1).

## Trait gradient analysis

To investigate variations of leaf N and P stoichiometry within vs across species, a trait gradient analysis was performed, following refs. 38–40. We fitted ordinary least squares (OLS) regressions to species-specific relationships between individual, species-level data points of leaf N (P, N:P) and the site-mean leaf N (P, N:P) of the site belonging to the respective data record. That is, a data point $x_{ij}$ recorded from species $i$ at site $j$ was regressed against $x_j$, the site-mean leaf N (P, N:P) for site $j$. Note that site-mean values are aggregated from data of all species sampled at the respective site. The regression slope ($b_i$) of the regression of $x_{ij}$ vs $x_j$ indicates the degree of intra-specific plasticity across sites with different environmental conditions, characterized by the site-level mean leaf N, P, and N:P, respectively[40]. A regression slope of zero can be interpreted as a non-plastic behavior (Fig. 1). A regression slope of one indicates an average and a regression slope >1 an above-average plastic response of leaf nutrient concentrations and stoichiometry within a species across sites with different community-mean concentrations and stoichiometry. We also quantified the species-specific range along the horizontal axis ($R_i$), given by values $x_j$. We defined ranges as the difference between 1% and 99% quantiles to reduce the influence of outliers. Species-specific range values were normalized by the overall range of site-level means $x_j$ to make ranges comparable across the three different traits investigated here. Normalized ranges are interpreted here as an indicator of the range of site conditions under which the respective species occurs. For the TGA, we considered only data from species that were recorded in at least five sites and used only data from sites where at least five different species were sampled. This filtering retained 3385 data points from 372 species. For fitting linear regression models, all data were log-transformed before performing the trait gradient analysis to improve the normality of the residuals. Normalized ranges were quantified based on the original, not log-transformed data.

We also conducted a "species variation decomposition", whereby we quantified and compared the coefficients of determination from comparing observed (unmodified) leaf N, P, and N:P values with modified values. Modifications followed two alternative assumptions, reflecting hypotheses in Fig. 1. The first assumes that variations arise only across species (no within-species variations), and values $x_{i,k}$ of species $i$ and observation $k$ are replaced by the respective species' mean, $\bar{x}_i$. The second assumes that variations arise only within species and each record is "normalized" such that the resulting species mean is equal to the global mean $\bar{x}$.

## Reporting summary

Further information on research design is available in the Nature Portfolio Reporting Summary linked to this article.

# Data availability

The records of leaf N and P concentrations in the current dataset were extended from Tian et al.[82] and combined with Sardans et al.[21]. As an

update, 45 environmental variables were added to each site in the dataset. The full dataset, along with code for complementing it with environmental covariates is published on Zenodo (https://doi.org/10.5281/zenodo.11071944)[109]. Source data for published figures are provided with this paper.

## Code availability

Code for the analysis is published on Zenodo (https://doi.org/10.5281/zenodo.11071816)[110].

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

## Acknowledgements

This work was supported by the National Key R&D Program of China (No. 2022YFD2201600, D.T.), grants from the National Natural Science Foundation of China (32271680, D.T.; 31800397, D.T.; and 31901086, Z.B.Y.) and the Strategic Priority Research Program of the Chinese Academy of Science (No. XDA26050401, D.T.). D.T. was supported by the Swiss Government Excellence Scholarship (2020-2021), the Young Elite Scientists Sponsorship Program by the China Association for Science and Technology (2021-2023, No. 2021QNRC001), and 5·5 Engineering Research & Innovation Team Project of Beijing Forestry University (No: BLRC2023A01). B.D.S. was funded by the Swiss National Science Foundation Grant (No. PCEFP2_181115). B.S. was funded by the University Research Priority Program "Global Change and Biodiversity" of the University of Zurich. This work was also supported by the TRY initiative on plant traits (http://www.try-db.org). The TRY database is hosted at the Max Planck Institute for Biogeochemistry (Jena, Germany) and supported by DIVERSITAS/Future Earth, the German Centre for Integrative Biodiversity Research (iDiv) Halle-Jena-Leipzig and EU project BACI (grant ID 640176). We would like to thank Lisha Lyu, Yaoqi Li, Ziyan Liao, Yunpeng Luo, and Mengya Liu for their helpful suggestions for data analysis. We acknowledge Zhiyao Tang, Wenxuan Han, Huifeng Hu, and the researchers who contributed their data in the global TRY database.

## Author contributions

D.T.: Conceptualization (lead); data curation (lead); funding acquisition (lead); project administration (lead); writing – original draft (lead); review and editing (lead). Z.B.Y.: Data curation (supporting); writing – original draft (equal); review and editing (equal). B.S.: Conceptualization (supporting); methodology (supporting); writing – review and editing (equal). J.K.: Data curation (supporting); review and editing (equal). J.Y.F.: Data curation (supporting); project administration (supporting); review and editing (equal); supervision (equal). B.D.S.: Conceptualization (equal); supervision (lead); software (lead); methodology (lead); writing – review and editing (equal).

## Competing interests

The authors declare no competing interests.
