## [Peer Review File · Nature Communications]

Environmental versus phylogenetic controls on leaf nitrogen and phosphorous concentrations of vascular plantsREVIEWER COMMENTS

Reviewer #1 (Remarks to the Author):

The present study appears to be an important work. However, it is uncertain whether it represents a substantial advancement in the field and if the methods employed sufficiently support the arguments made in this paper. Specific concerns are as follows. Please consider addressing these concerns to enhance the clarity and depth of your research findings in the paper.

Specific comments:

Lines 44-48: What is the biological significance of comparing Random Forest models, trait gradient analysis, and linear models? Do these comparisons reflect specific biological meanings or mechanisms?

Lines 90-92: This statement is difficult to comprehend. What is meant by "cannot be independent"? In the natural world, biological organisms and their environments are inherently adaptive processes, making complete independence challenging. Does it refer to the lack of independence between species identity and historical or current environmental factors? Linear mixed effect models can also analyze the interaction between species identity and environment.

Line 93: What about the interaction between environmental controls and species identity? It might not be an absolute dichotomy; their interactions could vary with changing environmental conditions or plant functional groups.

Lines 103-105: Why compare these three methods? Are there no better alternatives? Do the differences in these statistical methods hold any biological significance? Why compare variation within species and between species? Does within-species variation change with plant developmental stages, age, or individual size?

Figure 2: Are there issues of collinearity among the predictors? How does collinearity affect the results of the three methods? How many predictors are used in the models? If there are

numerous predictors, could the models' abilities to handle complex data differ, potentially affecting the results?

Lines 241-242: How can the comparability of variation within and between species be ensured?

Lines 249-250: What distinguishes "variation" from "plasticity"? What is the significance of within-species variations?

Lines 305-309: Linear mixed effect models can account for nonlinear relationships and interactions, although Random Forest models might be more effective. However, how do we interpret the meanings of Random Forest model parameters?

Reviewer #2 (Remarks to the Author):

The aims of this study are to disentangle the effects of community composition versus intra-specific variation on leaf N and P concentrations, and to quantify the impacts of environmental variation. This is important for many reasons and purposes, such as projections of future nutrient cycling in ecosystems and for vegetation model construction, specifically the definition of plastic and species-fixed components of leaf N and P. Given the central role of nutrient limitation for vegetation dynamics this is an important research question with wide-ranging potential impact.

In contrast to many previous studies the authors use non-linear random forest models (RF) for analysis, which is shown to greatly improve the models' predictive ability compared to traditional linear models (LM). The analysis convincingly (but diplomatically) debunks the "Biogeochemical Niche Hypothesis", i.e. that inter-specific variation was much more important than intra-specific. Finally, the authors describe very well the limitations of their analysis, e.g. that it doesn't separate the effects of individual acclimation and intra-specific genetic variation, which would be nice to do for modeling purposes. The text is very nicely written, and the figures are very accessible.

The only thing that I found slightly strange is the inclusion of modelled stomatal conductance and V_{cmax} as environmental predictors. These are plant physiological parameters. But because they are modelled here based on environmental factors, I guess they could be seen as derived aggregated environmental factors. Maybe it would be worth mentioning which environmental variables were used for their calculation? A grain of uncertainty for me is that I am not at all an expert in the statistical models used, so I cannot judge whether the RF model used is the best approach or if even more appropriate statistical methods are available. But as far as I can see, the methods used are very effective and quite innovative.

In summary, it looks like a brilliant paper.

Oskar Franklin

Reviewer #3 (Remarks to the Author):

Review of MS for Nature Comms

This ms reports on leaf nutrient contents (N, P N:P) along environmental gradients, asking what the relative contributions are of environmentally-driven variation within species, compared to species turnover. The question is important because leaf nutrients are important components of the Leaf Economic Spectrum that reflects a fundamental plant trade-off between acquisitive strategies for carbon assimilation and conservative strategies for long lasting leaf tissue. These traits are key to biomass productivity and are reflective of environment and changes in it.

As with the classic “nature vs nurture” argument, the background here is that opposing viewpoints may propose either (1) that species have fixed traits and that variation among sites is due to species turnover along environmental gradients. Contrastingly, the hypothesis is (2) that species are highly plastic and that observed leaf nutrients reflect environmental drivers within species. Arguments and evidence have been presented for both in the

literature. Here the authors attempt to resolve or measure the balance of evidence for each—their relative contributions—for leaf N & P. I like the idea, but I am concerned by some of the methodology, and I found that the presentation needs improvement.

I believe that the ms finds that both potential explanations have some support. Yet I found it quite difficult to glean an overall sense. This seems to be because: (a) methodology matters to the quantification; and (b) because in different places, the text variously emphasises one or other explanation, which may be a matter of expression; and (c) lastly because at least some of the motivation for the ms appears to be to test /falsify a claim that species have fixed chemistry, with negligible effect of environment within-species. So, I found myself confused, as other readers might be too.

To explain, the Title supports environment (by omitting species turnover). The results are reported in the Abstract as supporting an interpretation that Environment is of great/overriding importance. (there is a single line buried in the middle of the abstract stating similar effects, this is swamped by 7 successive lines emphasising environment). Then if I look at Fig 3 c,f,l, I have the impression species identity dominates. I hope that this indicates the problem.

The ms analyses a large, global dataset of leaf nutrients. Over 3,600 species across at >7500 sites and retaining at least 5 Site measurements. This is a strong data set covering climate space well.

I found the flow of the manuscript confused me as to the focus. The abstract, in particular, should be rewritten, because it appears that the topic of the paper was methodological comparison of two vastly different analytical methodologies. That is not the case that the authors had in mind I believe. I felt that the analytical comparison was subsidiary to the main question or environmentally driven plasticity vs species turnover. But I may be wrong.

A considerable part of the manuscript focusses on methodological concerns in the attribution of variance explanation between species and environment. Specifically, the authors draw attention to the case of random effects in mixed effects modelling. It is true

that the random effects soak up variation, some of which may be explained by fixed effects (covariates of interest). So, in a way of speaking, the random effects—species in this case—have priority. To work around this the authors use LM with species as a fixed effect either added before or after environment, to get a sense of the explained variation.

Trait-Gradient Analysis is an appealing methodology. But it has a flaw. It seeks to regress a trait measured on species *i* at site *j* on the community mean trait at site *j*. That species *i* must be part of the community at site *j* and contribute to the community mean trait. Therefore, the predictor and response are dependent. This dependence would be greater for more abundant species, for more widespread species and for less diverse communities. This makes the slope estimates unreliable, to an unknown degree.

The ms has potential to resolve some of the conflict in the literature about the relative importance of explanations of Leaf N&P. I hope that my comments here help in achieving that potential.

Specific comments

Title – this suggests (by omission) that variation across species is negligible.

L39 plasticity should be mentioned.

L43-44 emphasise this, it gets lost in following 2 sentences. Potentially you could append with “thus challenging claims that species turnover is paramount”

L44-48 these 2 sentences read as though they are contrasting patterns with respect to your research question (i.e. within spp environmentally-driven variation or species turnover). But it isn't, as far as I can tell. Suggest break into 2 sentences and introduce RF result with respect to species identity.

L48-51 yet this sentence appears to claim predominance of environment, contradicting earlier statement that both species and environment matter equally.

L51 ff. where is this in the results?

L76-79 this sentence does not seem to match the statement before about drivers.

L103-105 unclear how this methodological comparison relates to the question, rather than being a critique of a previous (paper using a) method.

L120 ff. this section is written without regard for the ecological question at hand. Highlight

the ecology, not the stats.

L136 can you specify that the recursive feature elimination applies in RF models? If indeed it does.

L140, and what are brown bars?

L141 as far as I am aware, t-values are not measure of effect size or magnitude, they are measures of signal/noise.

If predictor variables have all been centred and scaled by a SD, i.e., 'standardised', one can use the coefficient as a measure of effect size. Or in a nonlinear model one can simply use the predicted range (i.e. predicted max – predicted min) across a particular covariate.

L185-188 when you specify the species, if the species have restricted distributions, the name encodes something about the environment. it is not truly independent of environment.

L204-207 what does this mean? the order of calculation is unclear to me, and thus what it is meant to represent. Is it how much the trait varies among species at a site as a proportion of the site mean? Those kernels in Fig 4 e don't seem accurate, presumably a range of 0 is practically impossible. It means that either you have single species at a site, or the trait values are identical for the species at a site. Or am I missing something here?

Fig 3 c,f,i suggest species identity dominates. Yet Fig 4 f suggests practical equivalence of the relative contributions (40/50/60% across species). Where does Fig 4 f come from?

Fig 4 g,h,i these estimates should have uncertainty bars.

L238-245 this is really good. It is clear about the questions in conclusions, but I found that clarity hard to find elsewhere.

Supp Fig 1. Where does this result come from?

L486-> which results specifically Fig 3? Supp Fig 1?

L515-516 this is the dependency that I mentioned in general comments.

L535-542 this is unclear to me. Is this the method used to produce Supp Fig 1 or Fig 4f??

Also, the actual calculation here is underspecified. What is $x_{i,k}$? I think i is the species, j is the site, but what is k ?

Authors' response to reviewer's comments on the manuscript "*Widespread environment-driven plasticity in species' leaf nitrogen and phosphorus concentrations*" by Tian et al.

REVIEWER COMMENTS

Reviewer #1 (Remarks to the Author):

[Comment]: The present study appears to be an important work. However, it is uncertain whether it represents a substantial advancement in the field and if the methods employed sufficiently support the arguments made in this paper. Specific concerns are as follows. Please consider addressing these concerns to enhance the clarity and depth of your research findings in the paper.

[Reply]: We thank the reviewer for his or her appreciation of the importance of our work. We are convinced that our paper constitutes a substantial advancement in the field for the following reasons.

- Leaf N, P, and N:P are key traits for photosynthesis and biogeochemical cycling and our understanding of large-scale controls are directly relevant for informing global vegetation models. However, published findings appear to contradict each other. Recent publications have reported an almost exclusive control of leaf N, P, and N:P by phylogeny with a very minor role of environmental controls (Sardans *et al.* 2021; Vallicrosa *et al.* 2022). Yet, environmental variables are emphasized as important drivers for global patterns in another group of studies and used for producing globally upscaled maps of foliar nutrient concentrations (e.g., Han *et al.* 2005; Reich *et al.* 2010; Zhang *et al.* 2012; Maire *et al.* 2015; Asner *et al.* 2016; Tian *et al.* 2018; Tang *et al.* 2018; Dechant *et al.* 2023 Preprint).
- The methodological choices underlying these two groups of studies and implications for our understanding of the global ecology of foliar nutrient concentrations have never been addressed explicitly and their implicit conflict has not been discussed, let alone resolved in previous literature. Our results demonstrate that methodological choices underpin the conflicting conclusions of large-scale controls on leaf N, P, and N:P presented in previously published, high-profile studies (e.g., Sardans *et al.* 2021; Vallicrosa *et al.* 2022).
- We thus reconcile an implicit conflict and show a way forward, demonstrating the inherent limitation of linear mixed-effect models (LMM) for separating within vs across species variations in traits and demonstrating the power of more flexible, tree-based methods

(Random Forest, RF) for this modeling task. We quantify the relative contribution of species identity and environmental factors in driving variations in the data for leaf N, P, and N:P. These estimates contrast with the results by Sardans *et al.* 2021, published in Nature EE, and thus suggest that our understanding of dominating controls on these foliar traits must be revised.

- In addition to the RF analysis, we apply traits-gradient analysis for the first time to quantify the plasticity of global leaf N, P, and N:P within species and compare it among environmental settings and species. These are all novel insights that directly inform the fundamental question we address: what is the relative importance of species identity *vs* environment in controlling global variation in leaf N, P, and N:P?

The methods employed in our study support our arguments very directly in that the types of statistical models (LMM and RF) used in our study reflect the two main methods that have been used in the published literature on leaf N, P, and N:P modeling (e.g., Campetella *et al.* 2011; Bergmann *et al.* 2017; Mundim *et al.* 2021; Fajardo *et al.* 2022; see our reply in response to ‘**Lines 44-48:** What is the biological significance of comparing Random Forest models’ below). By contrasting their results, we are able to directly discuss an implicit conflict that was inherent in the published literature (using environment for global spatial modeling *vs* the claim that the environment is unimportant for leaf N, P, and N:P). (See also our reply to the reviewer comment on Lines 44-48).

Specific comments:

[Comment]: Lines 44-48: What is the biological significance of comparing Random Forest models, trait gradient analysis, and linear models? Do these comparisons reflect specific biological meanings or mechanisms?

[Reply]: We revised text in several places to clarify the relevance of these methods in the context of deciphering the power of environment *vs.* phylogenetic controls on leaf N and P concentrations of terrestrial plants. The revised text in the abstract now reads:

The substantial influence of environmental variables on within-species variations is identified using Random Forest models. In contrast, widely used linear-mixed effect models missed these effects almost completely. Our analysis demonstrates a substantial influence of the environment in driving plastic responses of leaf N, P, and N:P within species. This challenges previous reports of a fixed biogeochemical niche and an

overriding importance of species distributions in shaping global patterns of leaf N and P concentrations. (l. 46-52)

Two aspects are relevant for why we compared Random Forest models (RF) and linear (mixed-effect) models. First, these are the most widely used model types in the related literature. Linear mixed-effect modeling is widely established in the literature on plant traits (Mitchell *et al.* 2014; Knapp *et al.* 2016). In more recent years, the use of RF has gained traction (Rahman *et al.* 2021; Soltanikazemi *et al.* 2022) — presumably for their ease of use (minimal data pre-processing and hyperparameter tuning required) and strong performance (Cutler *et al.* 2007; Liu, Y 2014). Second, both model types can deal with data that contain both categorical (e.g., species identity) and continuous variables (e.g., environmental variables). In particular, linear mixed-effect models are a popular choice for dealing with structured data where structures introduce variations that remain unexplained by the fixed factors (e.g., variations ascribed to species identity) (Duursma *et al.* 2016; West *et al.* 2022). In that sense, yes, the model types relate to biological aspects of the data and modeling task at hand.

We added text in the Introduction to better explain this background (l. 88-105):

The conflicting attributions of observed variation in leaf N and P stoichiometry to phylogenetic *vs* environmental variables are related to an inherent methodological challenge. The separation of these variables is usually undermined by their lack of independence. The distribution of plant species is largely driven by the abiotic environment^{35,36}. Yet, species do occur over a certain range of environmental conditions. To what extent the environment drives phenotypic plasticity or genetic adaptation in leaf N and P stoichiometry also *within* species remains challenging to detect but is informative for testing the Biogeochemical Niche Hypothesis. Linear mixed-effect models have been widely employed for separating phylogenetic and environmental effects on leaf traits^{24,37}, motivated by their suitability to model structured data and their ability to control for phylogenetic effects and species identity as random terms, implicitly assuming that they are unrelated to the environment and given precedence over the latter in model fitting. More recently, tree-based statistical learning methods, for example Random Forest, have been shown to be suitable for modeling leaf N and P^{38,39}. These models, too, provide a natural way to simultaneously account for environmental (continuous) and phylogenetic (categorical) information. However, implications of methodological choices for separating environmental *vs* phylogenetic variables so far have not been explicitly considered.

[Comment]: Lines 90-92: This statement is difficult to comprehend. What is meant by "cannot be independent"? In the natural world, biological organisms and their environments are inherently adaptive processes, making complete independence challenging. Does it refer to the lack of independence between species identity and historical or current environmental factors? Linear mixed effect models can also analyze the interaction between species identity and environment.

[Reply]: Following your suggestion, we reformulated text here. As mentioned above (response to comment "Lines 44-48 ..."), relevant text now reads:

The separation of these variables is usually undermined by their lack of independence. The distribution of plant species is largely driven by the abiotic environment^{35,36}. Yet, species do occur over a certain range of environmental conditions and to what extent environment-driven plasticity or genetic adaptation drives variations in leaf N and P stoichiometry also *within* species remains challenging to detect but is informative for testing the Biogeochemical Niche Hypothesis. (l. 90-95)

[Comment]: Line 93: What about the interaction between environmental controls and species identity? It might not be an absolute dichotomy; their interactions could vary with changing environmental conditions or plant functional groups.

[Reply]: Thank you for this important point. We agree that there might not be an absolute dichotomy between environmental controls and species identity. We also agree that in addition to the non-orthogonal ("separation of these variables is usually undermined by their lack of independence") main effects of phylogeny and environment there could be phylogeny-by-environment interactions, i.e. different reaction norms of species to environmental variables. Indeed, that's what we find using the trait gradient analysis. We revised text here. It now reads (l. 106-111):

In view of these conflicting reports and methodological challenges, the question arises to what extent large-scale leaf N and P stoichiometric patterns are a reflection of different species (with their relatively fixed leaf nutrient stoichiometry) occurring at different sites along environmental gradients, and to what extent plasticity and genetic adaption, driven by the environment, drive variation within species and contribute to large-scale patterns of leaf nutrient stoichiometry (Fig. 1).

[Comment]: Lines 103-105: Why compare these three methods? Are there no better alternatives? Do the differences in these statistical methods hold any biological significance?

[Reply]: We replied to this point above (see point Lines 44-48 ...). As described in the Introduction section, previous contrasting viewpoints regarding the importance of phylogenetic vs environmental variables in shaping stoichiometric patterns in leaf N and P concentrations and N:P ratios are based on conflicting results from the linear regression model (LM) and linear mixed-effect models (LMMs). Compared with traditional LM and LMMs, RF have advantages including better performance for large datasets, ability to model complex interactions among predictor variables, learn non-linear effects of multiple environmental variables more effectively, and so on. Hence, we compared the power of the LM, LMMs and RF in explaining different components of variations in leaf N and P stoichiometry including variations across sites, within species, and across species. Our results highlight the importance of choosing suitable statistical models for revealing within-species variations and for properly reflecting the influence of environmental variables on interspecific and intraspecific variations of leaf N and P stoichiometry. We believe that differences in these statistical methods hold important biological significance in quantifying roles of phylogeny and environment on plant nutrient stoichiometry which implies diverse plant species' adaptation strategies during long-term evolution processes and their responses (especially trait plasticity) to current environmental changes. These biological significances are now discussed on lines 359-375.

[Comment]: Why compare variation within species and between species?

[Reply]: We compared variation within species and between species to answer the key scientific question here. We revised text in the Introduction to explain how the identification of variation within species is informative for our research questions.

Text on l. 81-84 reads:

[The “Biogeochemical Niche Hypothesis”] posits that each species is characterized by a fixed leaf stoichiometry (low within-species variability) [...]

Text on l. 93-95 reads:

To what extent the environment drives phenotypic plasticity or genetic adaptation drives in leaf N and P stoichiometry also *within* species remains challenging to detect but is informative for testing the Biogeochemical Niche Hypothesis.

[Comment]: Does within-species variation change with plant developmental stages, age, or individual size?

[Reply]: To obtain empirical evidence for a clear effect of developmental stage, age, or size on within-species variation of leaf N and P concentrations for global terrestrial plants would need huge projects in the future, which is out of the scope of our current dataset. Here, we had to implicitly assume that our data have no strong sampling biases with respect to plant developmental stage, age, or size. Because we did not find these variables coded together with the variables used in our analysis, we could unfortunately not test for such biases nor remove potential variation caused by plant developmental stage, age, or size (which might have reduced residuals and increased the strength of observed effects). The implicit assumption that original studies sampled plants of standardized developmental stage, age, and size across phylogeny and environment are typical for published large-scale analyses of leaf N, P, and N:P (e.g., Han *et al.* 2005; Reich *et al.* 2010; Zhang *et al.* 2012; Maire *et al.* 2015; Asner *et al.* 2016; Tian *et al.* 2018; Sardans *et al.* 2021; Vallicrosa *et al.* 2022).

To improve the global data set quality to the greatest extent, we included only those records of paired N and P concentrations of green and ripe leaves with detailed location information and excluded all records without site information or with unpaired N-P records. For the data collected via field investigation, we confirmed that all plant samples were harvested during the growing season and the duplicated records were carefully removed. Field sampling was primarily completed from July to August (details reported in Tian *et al.* 2019). Considering the consolidated sampling methods, large number of records and wide spatial distribution, we believe that our database here provides a useful and credible tool for the study of leaf nutrient stoichiometry.

[Comment]: **Figure 2:** Are there issues of collinearity among the predictors? How does collinearity affect the results of the three methods? How many predictors are used in the models? If there are numerous predictors, could the models' abilities to handle complex data differ, potentially affecting the results?

[Reply]: We designed our analysis with a strong view to mitigating the influence of collinearity. This is extensively reported (see Results, first subsection 'Variable selection and effects' and Fig. 2). The purpose of Fig. 2 is indeed to show that a relatively small subset of predictors is sufficient for modeling and that additional predictors do not add information — pointing to collinearity among them. A computationally intensive recursive feature elimination with a

cross-validation at each step was performed. The final selection of seven predictors for leaf N and nine predictors for leaf P and N:P corresponds to a complementary set of environmental variables, including climatic and edaphic, CO₂ and N-deposition. The identified important predictors are briefly discussed in the context of reports in the literature on l. 384-392:

Global mean atmospheric CO₂ was among the most important selected variables for all three target variables and the single most important predictor for leaf N:P. The direction of the effect on leaf N identified here (decline with increasing CO₂) is consistent with observations from Free Air CO₂ Enrichment (FACE) experiments^{71,74,80,81} and with the mechanistic understanding of the influence of CO₂ on N demand⁸².

And on l. 288-293:

Among the strongest effects identified here was a positive effect of N-deposition on leaf N (consistent with refs. 52-56) and N:P (consistent with refs. 10,57-59), a negative effect of the temperature of the coldest month on leaf P, and a positive effect of the same on leaf N:P (consistent with refs. 26,30,60). These apparently robust trait–environment relationships have given rise to spatial upscaling^{25,61}, mapping leaf traits with global coverage.

We are therefore confident that our approach effectively mitigates collinearity and associated risks of model overfitting, and that the selected factors represent a set of robustly identified drivers of leaf N, P, and N:P variations.

[Comment]: Lines 241-242: How can the comparability of variation within and between species be ensured?

[Reply]: In quantitative terms, the separation of variation within and across species is based on comparing the variance of species means with the pooled variance within species as described on l. 574-581 (“species variation decomposition”). The physical units (g N (P) g DM⁻¹)² of the two variances within and across species are therefore identical and thus the coefficients of determination calculated from the two types of modifications (the first replacing individual values with the corresponding species means and the second replacing individual values with the deviations from their species mean) can be compared. In terms of ecological meaning, comparing variations across *vs.* within species is interpretable with respect to fundamental questions related to the leaf economics spectrum, trait–environment relationships, or the role of phylogeny and other plant classifications — as demonstrated by a rich literature (e.g.,

Anderegg *et al.* 2018; Umaña *et al.* 2019; Fyllas *et al.* 2020). We added a cross-reference to the Methods section in the caption of Fig. 4 where results from the species variation decomposition are displayed (panel f).

[Comment]: Lines 249-250: What distinguishes "variation" from "plasticity"? What is the significance of within-species variations?

[Reply]: The term *plasticity*, or *phenotypic plasticity*, is commonly used to refer to the variability of a trait in response to environmental stimuli or the capacity of an organism to alter its phenotype in response to varying environmental conditions (Valladares *et al.* 2014; Adu *et al.* 2022). It is a biological phenomenon. In contrast, *variation* relates purely to the data. The relationship between within-species variations and environmental variables are interpreted here as an indicator of phenotypic plasticity or genetic adaptation of populations within a species (Stotz *et al.* 2021).

To better explain the significance of within-species variation for our research question, we modified text in several places.

On l. 81-84:

[The “Biogeochemical Niche Hypothesis”] posits that each species is characterized by a fixed leaf stoichiometry (low within-species variability), [...]

On l. 93-95:

To what extent the environment drives phenotypic plasticity or genetic adaptation in leaf N and P stoichiometry also *within* species remains challenging to detect but is informative for testing the Biogeochemical Niche Hypothesis.

[Comment]: Lines 305-309: Linear mixed effect models can account for nonlinear relationships and interactions, although Random Forest models might be more effective. However, how do we interpret the meanings of Random Forest model parameters?

[Reply]: We thank the reviewer for pointing out that mixed-effect models may indeed be formulated to deal with non-linear and interactive responses. We revised text to be more precise in our critique, which pertains specifically to how LMMs were used in the published literature on leaf N, P, and N:P (l. 310-317):

The large shared effect between environment and species identity (Supplementary Fig. 1) cannot be decomposed without relying on targeted experimental designs and their interactions and non-linearities are not considered in published LMM-based analyses^{20,24}. Our LMM model specification imitated their methodological choices. By design, in their LMMs the shared effect gets attributed to species identity if used as random-effects term, rather than to environmental variables used as fixed-effects terms in LMMs.

We added a sentence on l. 325-327 pointing to the limitation regarding interpretability of fitted model parameters in Random Forest analysis.

However, fitted model coefficients of RF are not always directly interpretable — in contrast to coefficients of LMMs.

Further research will be useful for diagnosing functional relationships from RF models fitted using machine learning. However, we consider this to be beyond the scope for the present paper which focuses on identifying effects based on their importance for out-of-sample predictions.

References cited above:

- Adu, M. O. et al. Root system traits contribute to variability and plasticity in response to phosphorus fertilization in 2 field-grown Sorghum [*Sorghum bicolor* (L.) Moench] cultivars. *Plant Phenomics* 2022, 0002 (2022).
- Anderegg, L. D. et al. Within-species patterns challenge our understanding of the leaf economics spectrum. *Ecol. Lett.* 21, 734-744 (2018).
- Asner, G. P., Knapp, D. E., Anderson, C. B., Martin, R. E. & Vaughn, N. Large-scale climatic and geophysical controls on the leaf economics spectrum. *PNAS* 113, E4043-E4051 (2016).
- Bergmann, J., Ryo, M., Prati, D., Hempel, S. & Rillig, M. C. Root traits are more than analogues of leaf traits: the case for diaspore mass. *New Phytol.* 216, 1130-1139 (2017).
- Campetella, G. et al. Patterns of plant trait-environment relationships along a forest succession chronosequence. *Agr. Ecosyst. Environ.* 145, 38-48 (2011).
- Cutler, D. R. et al. Random forests for classification in ecology. *Ecology* 88, 2783-2792 (2007).
- Dechant, B. et al. Intercomparison of global foliar trait maps reveals fundamental differences and limitations of upscaling approaches. *EarthArXiv* 1-57 (2023).

- Duursma, R. & Powell, J. *Mixed-effects models*. Sydney: Hawkesbury Institute for the Environment (2016).
- Fajardo, A., Piper, F. I. & García-Cervigón, A. I. The intraspecific relationship between wood density, vessel diameter and other traits across environmental gradients. *Funct. Ecol.* 36, 1585-1598 (2022).
- Fyllas, N. M. et al. Functional trait variation among and within species and plant functional types in mountainous mediterranean forests. *Front. Plant Sci.* 11, 212 (2020).
- Han, W., Fang, J., Guo, D. & Zhang, Y. Leaf nitrogen and phosphorus stoichiometry across 753 terrestrial plant species in China. *New Phytol.* 168, 377-385 (2005).
- Knapp, S., Stadler, J., Harpke, A. & Klotz, S. Dispersal traits as indicators of vegetation dynamics in long-term old-field succession. *Ecol. Indic.* 65, 44-54 (2016).
- Liu, Y. Random forest algorithm in big data environment. *Computer modelling & new technologies* 18, 147-151 (2014).
- Maire, V. et al. Global effects of soil and climate on leaf photosynthetic traits and rates. *Glob. Ecol. Biogeogr.* 24, 706-717 (2015).
- Mitchell, R. M., & Bakker, J. D. Quantifying and comparing intraspecific functional trait variability: a case study with *Hypochaeris radicata*. *Funct. Ecol.* 28, 258-269 (2014).
- Mundim, F. M., Vieira-Neto, E. H., Alborn, H., & Bruna, E. M. Disentangling the influence of water limitation and simultaneous above and belowground herbivory on plant tolerance and resistance to stress. *J. Ecol.* 109, 2729-2739 (2021).
- Rahman, M. et al. Disentangling the role of competition, light interception, and functional traits in tree growth rate variation in South Asian tropical moist forests. *Forest Ecol. Manag.* 483, 118908 (2021).
- Reich, P. B. et al. Evidence of a general $2/3$ -power law of scaling leaf nitrogen to phosphorus among major plant groups and biomes. *Proc. Natl Acad. Sci. USA* 277, 877-883 (2010).
- Sardans, J. et al. Empirical support for the biogeochemical niche hypothesis in forest trees. *Nat. Ecol. Evol.* 5, 184-194 (2021).
- Soltanikazemi, M., Minaei, S., Shafizadeh-Moghadam, H. & Mahdavian, A. Field-scale estimation of sugarcane leaf nitrogen content using vegetation indices and spectral bands of Sentinel-2: Application of random forest and support vector regression. *Comput Electron Agr.* 200, 107130 (2022).

- Stotz, G. C., Salgado-Luarte, C., Escobedo, V. M., Valladares, F. & Gianoli, E. Global trends in phenotypic plasticity of plants. *Ecol. Lett.* 24, 2267-2281 (2021)
- Tang, Z. et al. Patterns of plant carbon, nitrogen, and phosphorus concentration in relation to productivity in China's terrestrial ecosystems. *Proc. Natl. Acad. Sci. USA* 115, 4033-4038 (2018).
- Tian, D. et al. A global database of paired leaf nitrogen and phosphorus concentrations of terrestrial plants. *Ecology* 9, e02812 (2019).
- Tian, D. et al. Global leaf nitrogen and phosphorus stoichiometry and their scaling exponent. *Natl. Sci. Rev.* 5, 738-739 (2018).
- Umaña, M. N. & Swenson, N. G. Does trait variation within broadly distributed species mirror patterns across species? A case study in Puerto Rico. *Ecology*. 100, e02745 (2019).
- Valladares, F. et al. The effects of phenotypic plasticity and local adaptation on forecasts of species range shifts under climate change. *Ecol. Lett.* 17, 1351-1364 (2014).
- Vallicrosa, H. et al. Global maps and factors driving forest foliar elemental composition: the importance of evolutionary history. *New Phytol.* 233, 169-181 (2022).
- West, B. T., Welch, K. B. & Galecki, A. T. *Linear mixed models: a practical guide using statistical software.* Crc Press (2022).
- Zhang, S. B., Zhang, J. L., Slik, J. & Cao, K. F. Leaf element concentrations of terrestrial plants across China are influenced by taxonomy and the environment. *Glob. Ecol. Biogeogr.* 21, 809-818 (2012).

Reviewer #2 (Remarks to the Author):

[Comment]: The aims of this study are to disentangle the effects of community composition versus intra-specific variation on leaf N and P concentrations, and to quantify the impacts of environmental variation. This is important for many reasons and purposes, such as projections of future nutrient cycling in ecosystems and for vegetation model construction, specifically the definition of plastic and species-fixed components of leaf N and P. Given the central role of nutrient limitation for vegetation dynamics this is an important research question with wide-ranging potential impact.

In contrast to many previous studies the authors use non-linear random forest models (RF) for analysis, which is shown to greatly improve the models' predictive ability compared to

traditional linear models (LM). The analysis convincingly (but diplomatically) debunks the “Biogeochemical Niche Hypothesis”, i.e. that inter-specific variation was much more important than intra-specific. Finally, the authors describe very well the limitations of their analysis, e.g. that it doesn’t separate the effects of individual acclimation and intra-specific genetic variation, which would be nice to do for modeling purposes. The text is very nicely written, and the figures are very accessible.

The only thing that I found slightly strange is the inclusion of modelled stomatal conductance and V_{cmax} as environmental predictors. These are plant physiological parameters. But because they are modelled here based on environmental factors, I guess they could be seen as derived aggregated environmental factors. Maybe it would be worth mentioning which environmental variables were used for their calculation? A grain of uncertainty for me is that I am not at all an expert in the statistical models used, so I cannot judge whether the RF model used is the best approach or if even more appropriate statistical methods are available. But as far as I can see, the methods used are very effective and quite innovative.

In summary, it looks like a brilliant paper.

Oskar Franklin

[Reply]: We thank the reviewer for the appreciation of the value of our research.

Regarding the derived environmental variables ($V_{\text{cmax}25}$, $J_{\text{max}25}$, g_s), we added justification for the use of these as reflections of climate input variables. The complemented text in the Methods section (l. 459-463) reads:

Due to the intrinsic⁹⁷ and widely observed⁹⁸ link between $V_{\text{cmax}25}$, Rubisco, and leaf N and P contents, we used these estimates here as predictors for leaf N, P, and N:P, thus attempting to account for simultaneous effects of multiple climate variables (air temperature, VPD, elevation, irradiance, CO_2) on leaf nutrient stoichiometry.

We initially hypothesized that these simulated quantities effectively subsume the climatic information relevant for modeling leaf N, P, and N:P. However, the feature elimination did not retain these variables and they were not used for all subsequent analyses presented in Figs. 3 and 4. Therefore, we did not include a more extensive discussion of this result in the paper (also given the space constraints for the text).

Regarding the choice of RF as a statistical modelling tool, we now provide more information for justifying it (also in response to Reviewer #1). Text on l. 88-105 reads:

The conflicting attributions of observed variation in leaf N and P stoichiometry to phylogenetic vs environmental variables are related to an inherent methodological challenge. The separation of these variables is usually undermined by their lack of independence. The distribution of plant species is largely driven by the abiotic environment^{35,36}. Yet, species do occur over a certain range of environmental conditions. To what extent the environment drives phenotypic plasticity or genetic adaptation in leaf N and P stoichiometry also within species remains challenging to detect but is informative for testing the Biogeochemical Niche Hypothesis. Linear mixed-effect models have been widely employed for separating phylogenetic and environmental effects on leaf traits^{24,37}, motivated by their suitability to model structured data and their ability to control for phylogenetic effects and species identity as random terms, implicitly assuming that they are unrelated to the environment and given precedence over the latter in model fitting. More recently, tree-based statistical learning methods, for example Random Forest methods, have been shown to be suitable for modelling leaf N and P^{38,39}. These models, too, provide a natural way to simultaneously account for environmental (continuous) and phylogenetic (categorical) information. However, implications of methodological choices for separating environmental vs phylogenetic variables so far have not been explicitly considered.

Reviewer #3 (Remarks to the Author):

Review of MS for Nature Comms

[Comment]: This ms reports on leaf nutrient contents (N, P N:P) along environmental gradients, asking what the relative contributions are of environmentally-driven variation within species, compared to species turnover. The question is important because leaf nutrients are important components of the Leaf Economic Spectrum that reflects a fundamental plant trade-off between acquisitive strategies for carbon assimilation and conservative strategies for long lasting leaf tissue. These traits are key to biomass productivity and are reflective of the environment and changes in it.

As with the classic “nature vs nurture” argument, the background here is that opposing viewpoints may propose either (1) that species have fixed traits and that variation among sites

is due to species turnover along environmental gradients. Contrastingly, the hypothesis is (2) that species are highly plastic and that observed leaf nutrients reflect environmental drivers within species. Arguments and evidence have been presented for both in the literature. Here the authors attempt to resolve or measure the balance of evidence for each—their relative contributions—for leaf N & P. I like the idea, but I am concerned by some of the methodology, and I found that the presentation needs improvement.

[Reply]: We appreciate the very valuable feedback that helped us to improve the manuscript. Indeed, our research can be framed as an attempt to balance evidence and quantify relative contributions for contrasting interpretations. We have now adopted this “spot-on” formulation in the abstract (l. 39-40):

Here, we balance these contrasting views via measuring their relative contributions to the global leaf N/P stoichiometric pattern.

[Comment]: I believe that the ms finds that both potential explanations have some support. Yet I found it quite difficult to glean an overall sense. This seems to be because: (a) methodology matters to the quantification; and (b) because in different places, the text variously emphasises one or other explanation, which may be a matter of expression; and (c) lastly because at least some of the motivation for the ms appears to be to test /falsify a claim that species have fixed chemistry, with negligible effect of environment within-species. So, I found myself confused, as other readers might be too.

To explain, the Title supports environment (by omitting species turnover). The results are reported in the Abstract as supporting an interpretation that Environment is of great/overriding importance. (there is a single line buried in the middle of the abstract stating similar effects, this is swamped by 7 successive lines emphasising environment). Then if I look at Fig 3 c,f,I, I have the impression species identity dominates. I hope that this indicates the problem.

[Reply]: Indeed, our framing was initially guided strongly by the testing of the Biogeochemical Niche Hypothesis. To more accurately reflect our finding that within- and across-species variations contribute about equally to overall variation in the data and the fact that we quantify their relative contributions and the influence of the environment on each, we revised text in several places. First, we changed the title to:

Environmental versus phylogenetic controls on leaf nitrogen and phosphorous concentrations of land plants

Second, we revised the abstract (l. 36-52):

Global patterns of leaf nitrogen (N) and phosphorus (P) stoichiometry have been variably interpreted as reflecting phenotypic plasticity in response to the environment, or an overriding effect of the distribution of species growing in their biogeochemical niche. Here, we balance these contrasting views via measuring their relative contributions to the global leaf N/P stoichiometric pattern. We completed a comprehensive global data set and investigate how species identity and environmental variables control variation in leaf N, P, and N:P. We find that within-species variation contributes around half to the total variation in these three leaf variables, 29%, 31%, and 22% of which, respectively, are explained by environmental variables. Within-species plasticity along environmental gradients is highest for leaf N:P and lowest for leaf N and varies across species. The substantial influence of environmental factors on within-species variations was identified by Random Forest models. In contrast, widely used linear mixed-effect models missed these effect almost completely. Our analysis demonstrates a substantial influence of the environment in driving plastic responses of leaf N, P, and N:P within species. This challenges previous reports of a fixed biogeochemical niche and an overriding importance of species distributions in shaping global leaf patterns of N and P concentrations.

Third, we complemented text in the Discussion to better explain that the environmental variables influence both species distribution and leaf N, P, and N:P variations within species.

The text on l. 257-266 now reads:

The influence of the environment as a driver of leaf N, P, and N:P is demonstrated by three results presented here. First, environmental variables explain 30–45% of variation in community-weighted means across sites (Fig. 3a, d, g). This reflects the environmental filtering of species occurrence across environmental gradients. Second, besides species distribution, the environment influences leaf N, P, and N:P directly, driving variation within species. Random Forest models explained around 20–30% of this variation, as shown in Figure 3b, e, and h. Third, a large proportion of species expressed strong plasticity in leaf N, P, and N:P, whereby within-species variation paralleled across-site variation. This was expressed by the slope of species-specific regression lines in the trait gradient analysis (Fig. 4d).

[Comment]: The ms analyses a large, global dataset of leaf nutrients. Over 3,600 species across at >7500 sites and retaining at least 5 Site measurements. This is a strong data set covering climate space well.

I found the flow of the manuscript confused me as to the focus. The abstract, in particular, should be rewritten, because it appears that the topic of the paper was methodological comparison of two vastly different analytical methodologies. That is not the case that the authors had in mind I believe. I felt that the analytical comparison was subsidiary to the main question of environmentally driven plasticity vs species turnover. But I may be wrong.

[Reply]: The scientific question (controls on leaf N and P stoichiometry) and methodological choices are intertwined. We now revised text across the whole manuscript to put the science question in the foreground and to provide a better explanation of how finding answers to the science question is linked to methodological choices. Mentioning (and explaining) the link to the methods is necessary for discussing our results in the context of the literature. With this in mind, we re-wrote the abstract and added a paragraph in the Introduction that explains the link to methodological choices (l. 88-90):

The conflicting attributions of observed variation in leaf N and P stoichiometry to phylogenetic vs environmental variables are related to an inherent methodological challenge. [...]

[Comment]: A considerable part of the manuscript focuses on methodological concerns in the attribution of variance explanation between species and environment. Specifically, the authors draw attention to the case of random effects in mixed effects modeling. It is true that the random effects soak up variation, some of which may be explained by fixed effects (covariates of interest). So, in a way of speaking, the random effects—species in this case—have priority. To work around this the authors use LM with species as a fixed effect either added before or after environment, to get a sense of the explained variation.

Trait-Gradient Analysis is an appealing methodology. But it has a flaw. It seeks to regress a trait measured on species *i* at site *j* on the community mean trait at site *j*. That species *i* must be part of the community at site *j* and contribute to the community mean trait. Therefore, the predictor and response are dependent. This dependence would be greater for more abundant species, for more widespread species and for less diverse communities. This makes the slope estimates unreliable, to an unknown degree.

[Reply]: We don't consider the trait gradient analysis (TGA) as a statistical prediction method where the predictor and target variables, i.e., the variables plotted along the x and y-axis, must be independent. Across all observations, a correlation between individual observations and the site means emerges by design. The value of the TGA is in its information about the slope of species-specific regressions. A slope of zero would be indicative of a species-specific trait that does not vary within species. If indeed, this was an accurate model of the data-generating process, the TGA would identify it, as demonstrated below and in the supplementary material, uploaded along with our re-submission. A trait gradient analysis with synthetic data, generated by alternative processes that reflect the Biogeochemical Niche Hypothesis on the one hand and perfect stoichiometric plasticity on the other hand is shown in Fig. R1. A potential solution to the problem mentioned by the reviewer would be the use of jackknifing, i.e. leave the species being tested out of the calculation of the site mean. However, then each species would have a different x-axis for the calculation of its slope, making comparisons among slopes biased in a different way. We did not use such an approach because TGA in the literature has been conceived and applied in the way we used it here (e.g., Ackerly & Cornwell 2007; Kooyman *et al.* 2010; Gallagher & Leishman 2012; Cheng *et al.* 2016; Ottaviani *et al.* 2018; Dong *et al.* 2020; Zhou *et al.* 2022) and it would require a more thorough analysis to evaluate jackknifing in TGA.

Fig. R1: Trait gradient hypotheses demonstrated by applying the trait gradient method to synthetic data, generated with a data-generating process that followed the Biogeochemical Niche Hypothesis (left column) or the perfect stoichiometric plasticity assumption (right column).

The reviewer suggests that the “dependence would be greater for more abundant species, for more widespread species and for less diverse communities.” We tested whether the slopes show a dependence on abundance (number of observations per species in the data used for the TGA) and spread (number of sites at which a given species is observed). Slopes are defined at the species level. It is unclear how a dependence on community diversity could be tested. We found no relationship between the slope and species abundance or spread (see Fig. R2).

Fig. R2: Relationships between slopes in the trait gradient analysis and species abundance (top row) and spread (bottom row).

[Comment]: The ms has potential to resolve some of the conflict in the literature about the relative importance of explanations of Leaf N&P. I hope that my comments here help in achieving that potential.

[Reply]: Indeed, the comments by reviewer 3 were greatly helpful to improve our MS. Thank you very much.

Specific comments

[Comment]: Title – this suggests (by omission) that variation across species is negligible.

[Reply]: Thanks for your suggestion. We changed the title to:

Environmental versus phylogenetic controls on leaf nitrogen and phosphorous concentrations of land plants

[Comment]: L39 plasticity should be mentioned.

[Reply]: We revised this sentence. It now reads:

Global patterns of leaf nitrogen (N) and phosphorus (P) stoichiometry have been variably interpreted as reflecting phenotypic plasticity in response to the environment, or of an overriding effect by the global distribution of species growing in their biogeochemical niches [...] (1.36-39)

[Comment]: L43-44 emphasise this, it gets lost in following 2 sentences. Potentially you could append with “thus challenging claims that species turnover is paramount”

[Reply]: Thanks. We rewrote the Abstract completely as shown above.

[Comment]: L44-48 these 2 sentences read as though they are contrasting patterns with respect to your research question (i.e. within spp environmentally-driven variation or species turnover). But it isn't, as far as I can tell. Suggest break into 2 sentences and introduce RF result with respect to species identity.

[Reply]: Thanks. We rewrote the Abstract completely as shown above.

[Comment]: L48-51 yet this sentence appears to claim predominance of environment, contradicting earlier statement that both species and environment matter equally.

[Reply]: We rewrote the Abstract completely — also with a view to better reflect that influences of species identity and environmental variables contribute about equally.

[Comment]: L51 ff. where is this in the results?

[Reply]: We removed this statement in the revised Abstract.

[Comment]: L76-79 this sentence does not seem to match the statement before about drivers.

[Reply]: We removed this sentence and now write:

Overall, these contrasting viewpoints provide contentious interpretations for global leaf N and P stoichiometry variations and take conflicting viewpoints regarding the

importance of phylogenetic vs environmental controls in shaping these stoichiometric patterns.

[Comment]: L103-105 unclear how this methodological comparison relates to the question, rather than being a critique of a previous (paper using a) method.

[Reply]: We revised and added text in the Introduction to better explain the relevance of methodological choices in the context of the science question at hand (see also response to Reviewer 1). Most importantly, we added a paragraph in the Introduction to explain this (l. 88-105):

The conflicting attributions of observed variation in leaf N and P stoichiometry to phylogenetic vs environmental variables are related to an inherent methodological challenge. The separation of these variables is usually undermined by their lack of independence. The distribution of plant species is largely driven by the abiotic environment^{35,36}. Yet, species do occur over a certain range of environmental conditions. To what extent the environment drives phenotypic plasticity or genetic adaptation in leaf N and P stoichiometry also within species remains challenging to detect but is informative for testing the Biogeochemical Niche Hypothesis. Linear mixed-effect models have been widely employed for separating phylogenetic and environmental effects on leaf traits^{24,37}, motivated by their suitability to model structured data and their ability to control for phylogenetic effects and species identity as random terms, implicitly assuming that they are unrelated to the environment and given precedence over the latter in model fitting. More recently, tree-based statistical learning methods, for example Random Forest methods, have been shown to be suitable for modelling leaf N and P^{38,39}. These models, too, provide a natural way to simultaneously account for environmental (continuous) and phylogenetic (categorical) information. However, implications of methodological choices for separating environmental vs phylogenetic variables so far have not been explicitly considered.

[Comment]: L120 ff. this section is written without regard for the ecological question at hand. Highlight the ecology, not the stats.

[Reply]: We revised the text to better motivate the variable selection with regards to the ecological question at hand (l. 134-147):

We started by identifying the most important environmental variables separately for explaining variations in leaf N, P, and N:P. Reduced predictor sets, specific for leaf N, P, and N:P, respectively, enabled an improved model performance compared with models that included all 45 predictors (Fig. 2a-c) and were used for all subsequent analyses. In LMM models (Fig. 2d-f), N deposition (ndep) had the strongest effect on leaf N and leaf N:P variations within species (both positive). The temperature of the coldest month (tmonthmin) had the strongest effect on leaf P (negative). Atmospheric CO₂ concentrations (co2), ndep, and tmonthmin were among the most important predictors for leaf N, P, and N:P. Soil variables were only selected among the most important variables for leaf N (aluminum saturation of the soil solution, ALSA) and for leaf P (soil texture, measured by the water holding capacity class, AWC_CLASS).

[Comment]: L136 can you specify that the recursive feature elimination applies in RF models? If indeed it does.

[Reply]: Yes, the recursive feature elimination was done based on RF. We now provide this information in the caption of Figure 2.:

(a-c) Variable selection order determined by recursive feature elimination based on Random Forest and 5-fold cross-validation.

[Comment]: L140, and what are brown bars?

[Reply]: The brown bars in Figs. 2 a-c indicate the R^2 of a model with one additional predictor, as indicated by the label along the y-axis, added. As there is no clear gain when including additional predictors colored in brown, the final selection of variables is indicated by the green bars. In other words, the brown bars indicate the alternative environmental predictors that will not increase the R^2 of a model. We added text in the caption:

Brown bars indicate additional, next most important predictors which were not used for subsequent analyses.

[Comment]: L141 as far as I am aware, t -values are not measures of effect size or magnitude, they are measures of signal/noise. If predictor variables have all been centered and scaled by a SD, i.e., 'standardised', one can use the coefficient as a measure of effect size. Or in a nonlinear model one can simply use the predicted range (i.e. predicted max – predicted min) across a particular covariate.

[Reply]: We thank the reviewer for this insight. Although commonly used as measures for the variable importance, *t*-values cannot be interpreted as “effect magnitudes”. As suggested by the reviewer, we corrected this part of the analysis and describe it accordingly in the caption:

Effect magnitudes of the selected variables, measured by the coefficients of normalized fixed effects in LMMs.

[Comment]: L185-188 when you specify the species, if the species have restricted distributions, the name encodes something about the environment. it is not truly independent of environment.

[Reply]: We agree with you that plant species distribution encodes an influence of the environment and that they are not truly independent of the environment. We added text in the Introduction to clarify this upfront. On l. 88-92, we write:

The conflicting attributions of observed variation in leaf N and P stoichiometry to phylogenetic vs environmental variables are related to an inherent methodological challenge. The separation of these variables is undermined by their lack of independence. The distribution of plant species is largely driven by the abiotic environment^{35,36}.

[Comment]: L204-207 what does this mean? the order of calculation is unclear to me, and thus what it is meant to represent. Is it how much the trait varies among species at a site as a proportion of the site mean? Those kernels in Fig 4 e don't seem accurate, presumably a range of 0 is practically impossible. It means that either you have single species at a site, or the trait values are identical for the species at a site. Or am I missing something here?

[Reply]: In this paragraph, we report results on the extent of the distribution of species analyzed in the trait gradient analysis. The quantification of the species distribution range is defined here as the range of values of site-mean leaf N, P, and N:P at which a given species occurs. Of course, the species itself contributes to the site-mean. However, note that ...

“For the TGA, we considered only data from species that were recorded in at least five sites and used only data from sites where at least five different species were sampled.” (l.568-569).

The order of calculation is described in the Methods section (l. 563-573):

We defined ranges as the difference between 1% and 99% quantiles to reduce the influence of outliers. Species-specific range values were normalized by the overall range of site-level means x_j to make ranges comparable across the three different traits investigated here. Normalized

ranges are interpreted here as an indicator of the range of site conditions under which the respective species occurs. [...] Normalized ranges were quantified based on the original, not log-transformed data.

The kernel density plots are correct. They indicate a non-zero density around a range of zero. A clearer picture is provided by a histogram, indicating that there are observations in the lowest bin, but values themselves are not zero.

Fig. R3: Histogram of normalized ranges. The kernel density lines as shown in Fig. 4 of the manuscript are shown on top of histogram bars.

We chose to show kernel density plots instead of histograms as they enable data for all three traits to be plotted in the same panel, while enabling a clear visual distinction. Plotting histograms for the three traits on top of each other obscures visual clarity.

[Comment]: Fig 3 c,f,i suggest species identity dominates. Yet Fig 4 f suggests practical equivalence of the relative contributions (40/50/60% across species). Where does Fig 4 f come from?

[Reply]: Fig. 4f is titled as Species variation decomposition. We added a reference to Methods in the caption of Fig. 4. The respective section in the Method is on l. 574-581. Fig. 4f shows

how much of the variation arises within vs across species — irrespective of whether any of these variations are associated with environmental predictors and independent of any modeling. In contrast, Fig. 3 (c, f, i) shows how much of the variation environmental predictors explain and is based on modeling. We tried to make this distinction clear. For example, in the Abstract, we write:

We find that within-species variations contribute around half to the total variation in these three leaf variables, 29%, 31%, and 22% of which, respectively, are explained by environmental variables.

[Comment]: Fig 4 g,h,i these estimates should have uncertainty bars.

[Reply]: We added the 95% confidence intervals of the slope estimate as grey lines to the bar plots in the revision.

[Comment]: L238-245 this is really good. It is clear about the questions in conclusions, but I found that clarity hard to find elsewhere.

[Reply]: Thank you. We tried to clarify that “*Variation within species between sites is similarly large as variation among species*” also in other places of the manuscript.

[Comment]: Supp Fig 1. Where does this result come from?

[Reply]: We added a cross-reference to clarify this (green text below). In the Methods section, existing text reads:

Additionally, we separated individual and shared effects of site and species based on linear models where the respective predictors were fitted in different orders (Supp. Fig. 1).

Supp. Fig. 1 is referenced on l. 546.

[Comment]: L486-> which results specifically Fig 3? Supp Fig 1?

[Reply]: We are not sure what the reviewer refers to. Existing text on l. 486 reads:

“...variables alone by taking the difference between RF models fitted with environmental...”.

If we understand correctly, the reviewer asked the relationship between our results showed in Fig. 3 and Supp. Fig. 1 in the manuscript. We rewrote this part ‘Contrasting model performances’ to clarify our result in the revision. The results showed specifically in Fig 3 locates in line 165-178 which reads:

The Random Forest (RF) models, fitted to site-level aggregated data (mean across all observations by site), with the selected subset of environmental variables as predictors, achieved an R^2 of 0.46, 0.34, and 0.34 for leaf N, P, and N:P, respectively, in contrast to an R^2 of 0.17, 0.19, and 0.19 in LMs (Fig. 3 a,d,g). R^2_{margin} , measuring the proportion of variation explained by fixed (environmental) variables in LMMs, fitted to the full data, was only 0.04, 0.05, and 0.09 for leaf N, P, and N:P, respectively. In contrast, the proportions of variation explained by species identity were 0.68, 0.63, and 0.44 (ICC), respectively, for leaf N, P, and N:P ratio (Fig. 3 c,f,i). When RF models were fitted to the full data, environmental variables explained a larger proportion of the variation than they did in LMMs, namely 0.13, 0.26, and 0.16 vs 0.04, 0.05 and 0.09, respectively. A similar contrast in the predictive power of environmental variables in RF and LMs is seen with models fitted to the modified data that contained only within-species variation (Fig. 3 b,e,h). Here, RF models achieved an R^2 of 0.29, 0.31, and 0.22 for leaf N, P, and N:P, while LMs achieved an R^2 of 0.01, 0.03, and 0.07, respectively.

The results showed specifically in Supp. Fig. 1 locates in line 185-187 which reads:

Based on LMs, the shared effect of species identity and sites was dominant for leaf N, P, and N:P, explaining more than double of the variance explained by their separate effects (Supplementary Fig. 1).

[Comment]: L515-516 this is the dependency that I mentioned in general comments.

[Reply]: We replied to the point about the trait gradient analysis above and wrote that we found no relationship between the slope and species abundance or spread (see Fig. R2).

[Comment]: L535-542 this is unclear to me. Is this the method used to produce Supp Fig 1 or Fig 4f?? Also, the actual calculation here is underspecified. What is $x_{i,k}$? I think i the species, j is the site, but what is k ?

[Reply]: Yes, this is the method used to produce Fig 4f. We revised the wording in the revision as:

We also conducted a “species variation decomposition”, whereby we quantified and compared the coefficients of determination from comparing observed (unmodified) leaf N, P, and N:P values with modified values. Modifications followed two alternative assumptions, reflecting hypotheses in Fig. 1. The first assumes that variations arise only across species (no within-species variations), and values $x_{i,k}$ of species i and observation k are replaced by the respective species’ mean, \bar{x}_i . The second assumes that variations arise only within species and each record is “normalized” such that the resulting species mean is equal to the global mean \bar{x} .

References cited above:

- Ackerly, D. D & Cornwell, W. K. A trait-based approach to community assembly: partitioning of species trait values into within- and among-community components. *Ecol. Lett.* 10: 135-145.
- Cheng, J. H., Chu, P. F., Chen, D. M., Bai, Y. F. Functional correlations between specific leaf area and specific root length along a regional environmental gradient in Inner Mongolia grasslands. *Funct. Ecol.* 30: 985-997 (2016).
- Dong, N., et al. Components of leaf-trait variation along environmental gradients. *New Phytol.* 228: 82-94 (2020).
- Gallagher, R. V. & Leishman, M. R. Contrasting patterns of trait-based community assembly in lianas and trees from temperate Australia. *Oikos* 121: 2026-2035 (2012).
- Kooyman, R., Cornwell, W., Westoby, M. Plant functional traits in Australian subtropical rain forest: partitioning within-community from cross-landscape variation. *J. Ecol.* 98: 517-525 (2010).
- Ottaviani, G., Tsakalos, J. L., Keppel, G., Mucina, L. Quantifying the effects of ecological constraints on trait expression using novel trait-gradient analysis parameters. *Ecol. Evol.* 8: 435-440 (2018).
- Zhou, J. H., Cieraad, E., van Bodegom, P. M. Global analysis of trait–trait relationships within and between species. *New Phytol.* 233: 1643-1656 (2022).

REVIEWERS' COMMENTS

Reviewer #1 (Remarks to the Author):

Thank you for the thorough revisions the author made to address my concerns. The quality of the article has significantly improved. I believe the paper has now reached the standard for publication in Nature Communications.

Reviewer #3 (Remarks to the Author):

This work provides a balanced analysis of contrasting views for leaf chemical traits along environmental gradients concerning the relative roles of specie identity and turnover versus within species plasticity. I thank the authors for their consideration and response to my review. I found the ms much improved. it should be an important paper for our field. I have just a couple of comments (not necessarily to be acted upon) and few minor suggestions.

There is no doubt that flexible Machine learning methods such as Random Forests and Boosted Regression Trees are very useful. Perhaps most useful when exploring partial dependence plots to (a) identify most important predictors, (b) learn about the functional form of relationships, (c) identifying which interactions are most important. This is important for our field to understand how and why the RF models explain so much more than the LMMs and how do we improve LMMs, which being parametric offer advantages for interpretation and generality over machine learning methods. Perhaps a comment in the discussion?

Thanks for the work to address my comments about TGA. I accept your conclusions. I am still privately concerned about TGA. But I do not wish to impeded progress on publishing this work. Using your own code, I found that shortening the range of species greatly reduced the slope of $T_{i,k}$ on T_{bar_k} . intermediate simulations with a mix of niche turnover and plasticity produced a range of slopes. And I am still trying to make sense of them and cannot rid myself of the sense of circularity. Indeed, your simulation of Perfect plasticity encodes direct dependence of species trait observation on the site trait mean. Anyhow, this is not the concern of the current paper. You do find quite a strong degree of plasticity; on that we

can agree. Good luck with your important work. I enjoyed thinking about it.

Minor edits

L40 suggest “compiled” for “completed”

L250 suggest “compiled” for “completed”

L397ff can this be rewritten, something like this:

“The role of environment in both filtering species and influencing traits within species, combined with...”

Reviewer #3 (Remarks on code availability):

I installed and ran the supplementary code for the rejoinder. it was clear, commented well. I did not run the main scripts. Looking at the organisation of the other scripts, the good organisation and clear coding lead me to be fairly confident in them. running full analyses would take considerable time. I noted that the data are not all available in the Github repo indicated.

REVIEWER COMMENTS

Reviewer #1 (Remarks to the Author):

[Comment]: Thank you for the thorough revisions the author made to address my concerns. The quality of the article has significantly improved. I believe the paper has now reached the standard for publication in Nature Communications.

[Reply]: We thank the reviewer for his or her comments which give us opportunity to improve our manuscript.

Reviewer #3 (Remarks to the Author):

[Comment]: This work provides a balanced analysis of contrasting views for leaf chemical traits along environmental gradients concerning the relative roles of specie identity and turnover versus within species plasticity. I thank the authors for their consideration and response to my review. I found the ms much improved. it should be an important paper for our field. I have just a couple of comments (not necessarily to be acted upon) and few minor suggestions.

[Reply]: We thank the reviewer for his or her appreciation of the importance of our work.

[Comment]: There is no doubt that flexible Machine learning methods such as Random Forests and Boosted Regression Trees are very useful. Perhaps most useful when exploring partial dependence plots to (a) identify most important predictors, (b) learn about the functional form of relationships, (c) identifying which interactions are most important. This is important for our field to understand how and why the RF models explain so much more than the LMMs and how do we improve LMMs, which being parametric offer advantages for interpretation and generality over machine learning methods. Perhaps a comment in the discussion?

[Reply]: Thank you for this important suggestion. We agree with you about the advantage of RF. Hence, we mentioned this point in the Discussion as follows:

Random Forest models learn interactive and non-linear effects of multiple environmental variables more effectively, detect trait–environment relationships beyond those arising from species composition (Fig. 3b, e, h), and yield superior results compared with linear regression models in out-of-sample evaluations when fitted to leaf N and P data at hand (Fig. 3a, d, g). However, fitted model coefficients of RF are not always directly interpretable — in contrast to coefficients of LMMs. (l. 328-334).

In Fig. 2, we also provided information about feature importance and effect magnitudes and direction, although the feature importance order was established by feature elimination rather than by analysing a single fitted model, and the effect magnitude and direction was from LMMs, not the RF.

Specific comments:

[**Comment**]: Thanks for the work to address my comments about TGA. I accept your conclusions. I am still privately concerned about TGA. But I do not wish to impeded progress on publishing this work. Using your own code, I found that shortening the range of species greatly reduced the slope of $T_{i,k}$ on $T_{bar,k}$. intermediate simulations with a mix of niche turnover and plasticity produced a range of slopes. And I am still trying to make sense of them and cannot rid myself of the sense of circularity. Indeed, your simulation of Perfect plasticity encodes direct dependence of species trait observation on the site trait mean. Anyhow, this is not the concern of the current paper. You do find quite a strong degree of plasticity; on that we can agree. Good luck with your important work. I enjoyed thinking about it.

[**Reply**]: We are also interested in conducting further exploration about TGA in the next work. Thank you very much for your suggestion about this important point.

Minor edits:

[**Comment**]: L40 suggest “compiled” for “completed”

[**Reply**]: Revised according to this suggestion.

[**Comment**]: L250 suggest “compiled” for “completed”

[**Reply**]: Revised according to this suggestion.

[**Comment**]: L397ff can this be rewritten, something like this:

“The role of environment in both filtering species and influencing traits within species, combined with...”

[**Reply**]: This sentence was revised as follows:

The role of environment in both filtering species and influencing traits within species, combined with the widespread use of linear mixed effects models with limited function of distinguishing

statistical independence of phylogeny and environmental effects, predisposed published analyses to miss strong effects of environmental variables (l. 423-427).

Reviewer #3 (Remarks on code availability):

[Comment]: I installed and ran the supplementary code for the rejoinder. it was clear, commented well. I did not run the main scripts. Looking at the organisation of the other scripts, the good organisation and clear coding lead me to be fairly confident in them. running full analyses would take considerable time. I noted that the data are not all available in the Github repo indicated.

[Reply]: We thank the reviewer for his or her appreciation of the codes. All the codes and data are available at Github (https://github.com/stineb/leafnp_data) and Zenodo (<https://doi.org/10.5281/zenodo.11071944>). Source data are available at Zenodo (<https://zenodo.org/records/11105009>). In addition, figures can be re-created directly using intermediate data objects stored at GitHub (https://github.com/stineb/leafnp_data) in `data/data_published_figures`.